# Peptidomimetic inhibitors of TMPRSS2 block SARS-CoV-2 infection in cell culture

Lukas Wettstein [1,9], Philip Maximilian Knaff[2,3,9], Christian Kersten [4,9], Patrick Müller[4], Tatjana Weil [1], Carina Conzelmann[1], Janis A Müller [1,5], Maximilian Brückner[2,3], Markus Hoffmann [6,7], Stefan Pöhlmann [6], Tanja Schirmeister[4], Katharina Landfester[3], Jan Münch [1,8,10✉] & Volker Mailänder [2,3,10✉]

The transmembrane serine protease 2 (TMPRSS2) primes the SARS-CoV-2 Spike (S) protein for host cell entry and represents a promising target for COVID-19 therapy. Here we describe the in silico development and in vitro characterization of peptidomimetic TMPRSS2 inhibitors. Molecular docking studies identified peptidomimetic binders of the TMPRSS2 catalytic site, which were synthesized and coupled to an electrophilic serine trap. The compounds inhibit TMPRSS2 while demonstrating good off-target selectivity against selected coagulation proteases. Lead candidates are stable in blood serum and plasma for at least ten days. Finally, we show that selected peptidomimetics inhibit SARS-CoV-2 Spike-driven pseudovirus entry and authentic SARS-CoV-2 infection with comparable efficacy as camostat mesylate. The peptidomimetic TMPRSS2 inhibitors also prevent entry of recent SARS-CoV-2 variants of concern Delta and Omicron BA.1. In sum, our study reports antivirally active and stable TMPRSS2 inhibitors with prospects for further preclinical and clinical development as antiviral agents against SARS-CoV-2 and other TMPRSS2-dependent viruses.

[1] Institute of Molecular Virology, Ulm University Medical Center, 89081 Ulm, Germany. [2] Dermatology Clinic of the University Medicine of the Johannes Gutenberg University Mainz, 55131 Mainz, Germany. [3] Max Planck Institute for Polymer Research, 55128 Mainz, Germany. [4] Institute of Pharmaceutical and Biomedical Sciences, Johannes Gutenberg University Mainz, 55128 Mainz, Germany. [5] Institute of Virology, Philipps University Marburg, Marburg, Germany. [6] Infection Biology Unit, German Primate Center, 37077 Göttingen, Germany. [7] Faculty of Biology and Psychology, Georg-August-University Göttingen, 37073 Göttingen, Germany. [8] Core Facility Functional Peptidomics, Ulm University Medical Center, 89081 Ulm, Germany. [9] These authors contributed equally: Lukas Wettstein, Philip Maximilian Knaff, Christian Kersten. [10] These authors jointly supervised this work: Jan Münch, Volker Mailänder. ✉email: jan.muench@uni-ulm.de; mailaend@mpip-mainz.mpg.de

Severe acute respiratory syndrome coronavirus 2 (SARS-CoV-2) is the causative agent of coronavirus disease 2019 (COVID-19). By the end of March 2022, the WHO reported more than 470 million confirmed SARS-CoV-2 infections worldwide, resulting in more than 6 million deaths since its first occurrence in late 2019[1]. COVID-19 is characterized by a mild-to-moderate respiratory illness and infected individuals usually recover without requiring special treatment. Older people, and those with underlying medical problems such as cardiovascular disease, diabetes, chronic respiratory disorders, and cancer are more likely to develop serious illness, characterized by respiratory failure, shock, and multiorgan dysfunction[2–5]. SARS-CoV-2 is primarily transmitted through aerosols and droplets of saliva. The inhaled virus may then establish infection in epithelial cells of the upper respiratory tract from which it may further disseminate to lower airway epithelial and alveolar cells, and other organs such as the gastrointestinal tract or heart[6–9]. Measures to prevent or mitigate SARS-CoV-2 spread include lockdown strategies, social distancing, quarantining, use of face masks, and hygiene concepts. The implementation of effective SARS-CoV-2 vaccination programs are the best defense against COVID-19 and have raised hopes that the pandemic is nearing an end. However, the emergence of viral variants of concern that escape pre-existing immunity and are associated with increased transmissibility and higher case fatality rates, as well as the slow vaccine rollout in most countries, may compromise efforts to control the pandemic[10–14]. Currently, the antivirals nirmatrelvir (in combination with ritonavir for CYP-inhibition, Paxlovid®) and molnupiravir (Lagevrio®) are available for the treatment of COVID-19[15,16]. Nevertheless, it remains imperative to develop further potent therapeutic interventions for COVID-19 therapy.

SARS-CoV-2 is an enveloped, positive-sense single-stranded RNA virus. Infection is mediated by the viral spike (S) protein, a homotrimeric transmembrane glycoprotein. The spike glycoprotein is composed of S1 and S2 subdomains. The S1 subdomain encodes for the receptor-binding domain and is responsible for binding to angiotensin-converting enzyme 2 (ACE2), the primary receptor for SARS-CoV-2[17–19]. Subsequently, the transmembrane protease serine subtype 2 (TMPRSS2) primes the S protein which triggers conformational changes in S2 leading to fusion of the viral with the cellular membrane and delivery of the nucleocapsid into the cytoplasm[19]. Of note, TMPRSS2 not only cleaves and primes SARS-CoV-2 spike, but also surface proteins of several other viruses, including the hemagglutinin (HA) of certain influenza A virus strains, the fusion protein (F) of human metapneumovirus, and the spike proteins of human coronavirus 229E (HCoV-229E), Middle East respiratory syndrome coronavirus (MERS-CoV), and SARS-CoV[19–25]. TMPRSS2 priming is essential for triggering fusion of these virions with target cells and disruption of TMPRSS2 expression was found to markedly reduce influenza A virus, SARS-CoV and MERS-CoV infection and pathogenesis in mice. Importantly, TMPRSS2 knockout mice are phenotypically similar to wild-type animals, suggesting that the protease is not essential, rendering TMPRSS2 a very promising target for broad-spectrum antiviral agents[26–28].

TMPRSS2 belongs to the family of type II transmembrane serine proteases (TTSP) which control a variety of physiological processes, including epithelial differentiation, homeostasis, iron metabolism, hearing, and blood pressure regulation[29]. The family of TTSP comprises a total of 17 members in four subfamilies (matriptase, corin, hepsin/TMPRSS, and HAT/DESC subfamily) with a common domain structure including an intracellular N-terminus, a transmembrane domain which anchors the protease in the cell membrane, and an extracellular C-terminus harboring a serine-protease domain[30]. TMPRSS2 belongs to the hepsin/TMPRSS subfamily with a total of seven serine proteases including TMPRSS2–5, MSPL, hepsin, and enteropeptidase. The development of TMPRSS2 inhibitors is hampered by the fact that no crystal structure was available until early 2021. Up to date, only few substrate analog inhibitors of TMPRSS2 have been described[31,32]. Approved drugs that are known to inhibit TMPRSS2 may be suitable for off-label use and repurposing in COVID-19 prevention and therapy. Noteworthy are camostat mesylate (CM) and nafamostat, which are used for the treatment of chronic pancreatitis in Japan, as well as the endogenous protease inhibitor alpha-1 antitrypsin (α₁-AT), and the mucolytic cough suppressant bromhexine[19,33–35]. However, the selectivity of some of these inhibitors is low, little is known about structure-activity relationships and the therapeutic effect of CM in COVID-19 has not yet been reported.

Herein, we describe our development of peptidomimetic inhibitors for TMPRSS2 for the treatment of SARS-CoV-2 infection. Using computational modeling and docking of combinatorial peptide libraries, we identified high-scoring binders which were synthesized and characterized with respect to protease inhibition, selectivity, and antiviral activity. This approach allowed to identify lead candidates that efficiently inhibit TMPRSS2 enzyme activity and block SARS-CoV-2 spike-driven entry into target cells. Furthermore, we demonstrate that these peptidomimetic TMPRSS2 inhibitors prevent authentic SARS-CoV-2 infection, including the variants of concern Alpha and Beta. The tested peptidomimetics are stable in human plasma and serum for at least 10 days, suggesting that these TMPRSS2 inhibitors are promising leads for further development as antiviral drugs in COVID-19 therapy and other viral diseases.

## Results

**Structure-based design of TMPRSS2 inhibitors.** For the identification of peptide-based TMPRSS2 inhibitors as a potential treatment of SARS-CoV-2 infection, molecular docking studies were performed. As no crystal structure of TMPRSS2 was available in the protein data bank (PDB) at the beginning of the study, we employed a matriptase crystal structure to build a surrogate model[36,37]. Matriptase shares 41% sequence identity with TMPRSS2 and previously described substrate analog TMPRSS2 inhibitors showed no selectivity over matriptase[31]. Computational redocking of the crystallographic ligand and the matriptase surrogate yielded a ligand orientation comparable to that in the crystal structure (RMSD of 1.8 Å), thus validating the surrogate model (Supplementary Fig. 3a). In addition, a homology model of TMPRSS2 was built using hepsin (43% sequence identity) as a template. Both models were tested for selectivity by docking of known binders and non-binders. Subsequent receiver operating characteristic (ROC) analysis indicated reliable discrimination between known binders and decoys for both, the homology and the surrogate model (Supplementary Fig. 3b, c). Previous studies characterized the substrate requirements of TMPRSS2, revealing a preference for arginine in P1 position, a glycine or proline residue in P2 position, and a D-configured arginine in P3 position (Supplementary Table 1)[31]. Based on this substrate preference, a reference binder comprising a N-terminal acetyl cap and a C-terminal aldehyde serine trap with the sequence ace-D-Arg-Gly/Pro-Arg-aldehyde was designed and docked to the matriptase-based TMPRSS2 surrogate model (Fig. 1a and Supplementary Table 3) and the TMPRSS2 homology model (Fig. 1b). Our dockings show that the reference binder ace-D-Arg-Pro-Arg-aldehyde binds to the reactive center of the surrogate and the homology model. To optimize the binding affinity of the reference binder, we successively altered the residues at P1-P3 position using proteinogenic and non-proteinogenic amino acids and docked the resulting structures to both, the matriptase

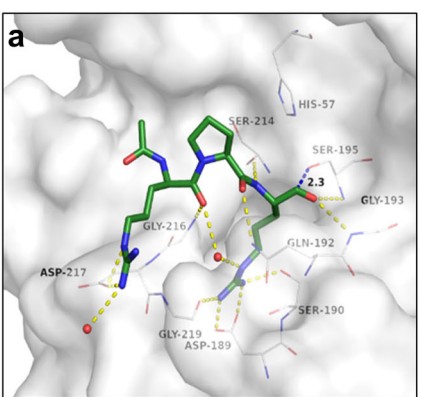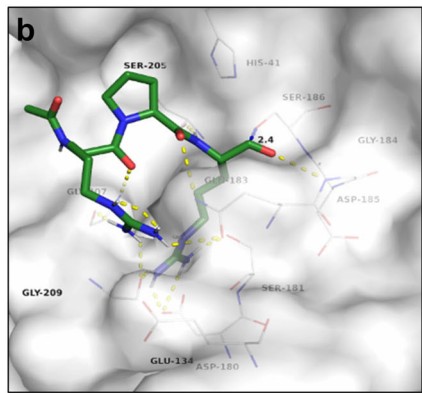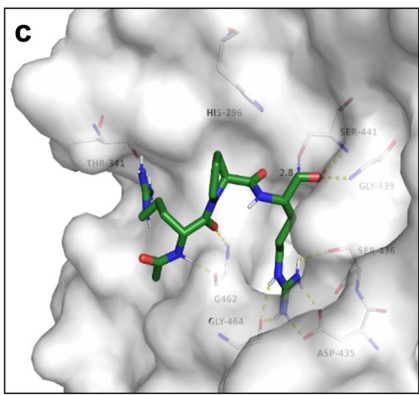

**Fig. 1 Predicted binding of reference binder. a** Docking of ace-D-Arg-Pro-Arg-aldehyde reference binder to matriptase surrogate model (white carbon atoms and surface). For a clear view, only residues forming polar interactions (yellow dashed lines), and the catalytic residues Ser-195 and His-57 are depicted. **b** Docking of ace-D-Arg-Pro-Arg-aldehyde reference binder to hepsin-based TMPRSS2 homology model (white carbon atoms and surface). For a clear view, only residues forming polar interactions (yellow dashed lines) and the catalytic residues Ser-186 and His-41 are depicted. **c** Docking of ace-D-Arg-Pro-Arg-aldehyde reference binder to TMPRSS2 crystal structure (PDB-ID: 7MEQ, white carbon atoms and surface). For a clear view, only residues forming polar interactions (yellow dashed lines), and the catalytic residues Ser-441 and His-296 are depicted. For all panels, carbon atoms of docked ligands are shown in green, oxygen in red, and nitrogen in blue. The distance between the nucleophilic serine oxygen and the electrophilic carbon atom of the serine trap in angstrom is illustrated by a dashed blue line.

**Table 1 Assembled peptidomimetic inhibitor library selected for synthesis.**

| Compound | N-cap | P3 | P2 | P1 | Serine trap |
|---|---|---|---|---|---|
| 1 | ace | D-Arg | Pro | Arg | kbt |
| 2 | ace | Arg | Pro | Arg | kbt |
| 3 | ace | D-His | Pro | Arg | kbt |
| 4 | ace | His | Pro | Arg | kbt |
| 5 | ace | Asn | Pro | Arg | kbt |
| 6 | ace | D-Arg | Pip | Arg | kbt |
| 7 | ace | D-Arg | Cyc | Arg | kbt |
| 8 | ace | D-Arg | Thr | Arg | kbt |

*ace* N-terminal acetyl cap, *Cyc* cyclobutylalanine, *Pip* pipecolinic acid, *kbt* ketobenzothiazole. The bond between the P1 position and serine trap is the site of nucleophilic attack by the protease catalytic triad.

surrogate and TMPRSS2 homology model. Compounds were then ranked based on their binding score (Supplementary Tables 2–4), the plausibility of their binding mode to the S1-S3 sub-pockets of the TMPRSS2 active site, and the proximity of the aldehyde serine trap to the catalytic Ser-186/195 (TMPRSS2 homology model/matriptase enumeration), as well as commercial availability of their building blocks. Overall, the D-configuration for P3 residue was favored to improve metabolic stability. The most promising compounds were chosen for solid-phase synthesis, whereby the aldehyde serine trap used for in silico modeling was exchanged by a well-established ketobenzothiazole, yielding a library of peptidomimetic inhibitors (Table 1 and Supplementary Fig. 4).

We retrospectively validated our molecular docking results by employing a TMPRSS2 crystal structure (PDB-ID: 7MEQ)[38] that became available during the course of this study. Superposition with the TMPRSS2 structure confirmed the accuracy of the homology- and the matriptase surrogate model, with overall $C_\alpha$-RMSD values of 0.6 Å for both models (Supplementary Fig. 5). The TMPRSS2 structure allowed successful redocking of its crystallographic ligand and enabled discrimination of binders and decoys (Supplementary Fig. 3d, e). Accordingly, the reference binder ace-D-Arg-Pro-Arg-aldehyde accommodated the substrate-binding pocket of the TMPRSS2 crystal structure, with the electrophilic serine trap being in close proximity to the

catalytic serine 441 (Fig. 1c). Retrospective docking studies of the selected peptide sequences (Table 1) against this crystal structure confirmed the predicted binding modes found in the homology model and matriptase surrogate and their classification as potential TMPRSS2 binders (Supplementary Tables 3 and 4).

**Designed peptidomimetic inhibitors block TMPRSS2 and matriptase activity.** We next investigated the impact of the peptidomimetic inhibitors on the activity of closely related matriptase and TMPRSS2 enzymes. To this end, the respective purified proteases were incubated with the compounds 1–8, followed by adding a protease-specific reporter substrate that allowed monitoring of protease activity over time. Overall, the compounds suppressed TMPRSS2 activity in the low nanomolar range ($K_i = 2.5–215.9$ nM) and inhibit matriptase with comparable activity (Fig. 2a, b and Table 2). Compound 1 which contains the peptide sequence of the reference binder showed an activity of $K_i = 86.7$ nM, while the compounds 2, 3, 4, 5, and 7 were most active against isolated TMPRSS2, with inhibitory constants of 2.5–57.5 nM. The compounds 2 ($K_i = 3.8$ nM) and 5 ($K_i = 2.5$ nM) were 2–3-fold more active than the active metabolite of CM, FOY-251 ($K_i = 9.7$ nM). Compounds 6 and 8 were the least active with $K_i$ values of 71.5 and 215.9 nM, respectively. The activity of the inhibitors against TMPRSS2 correlated with their activity against matriptase (Fig. 3). Yet, compound 1 showed the highest selectivity (~51-fold) for matriptase while FOY-251 showed the highest selectivity (~18-fold) for TMPRSS2.

For systemic administration of the protease inhibitors, high selectivity over off-target proteases is required to reduce side effects. To investigate potential interference of the peptidomimetic inhibitors with serine proteases involved in coagulation, we assessed their activity against thrombin and factor Xa (Fig. 2c, d and Table 2). All compounds excluding compound 1 displayed a >100-fold selectivity against thrombin (Table 2). Compound 1 showed no selectivity over factor Xa while compounds 2, 3, 4, 5, 6, and 7 revealed 1.6–38.9-fold selectivity compared to TMPRSS2. Truncation of the ketobenzothiazole serine trap moiety to a ketothiazole did not improve activity against matriptase, nor thrombin/factor Xa selectivity, and further reduction to an alcohol abolished antiprotease activity (Supplementary Fig. 6 and Supplementary Table 5). Considering the inhibitory constants and selectivity over potential off-target coagulation proteases, the

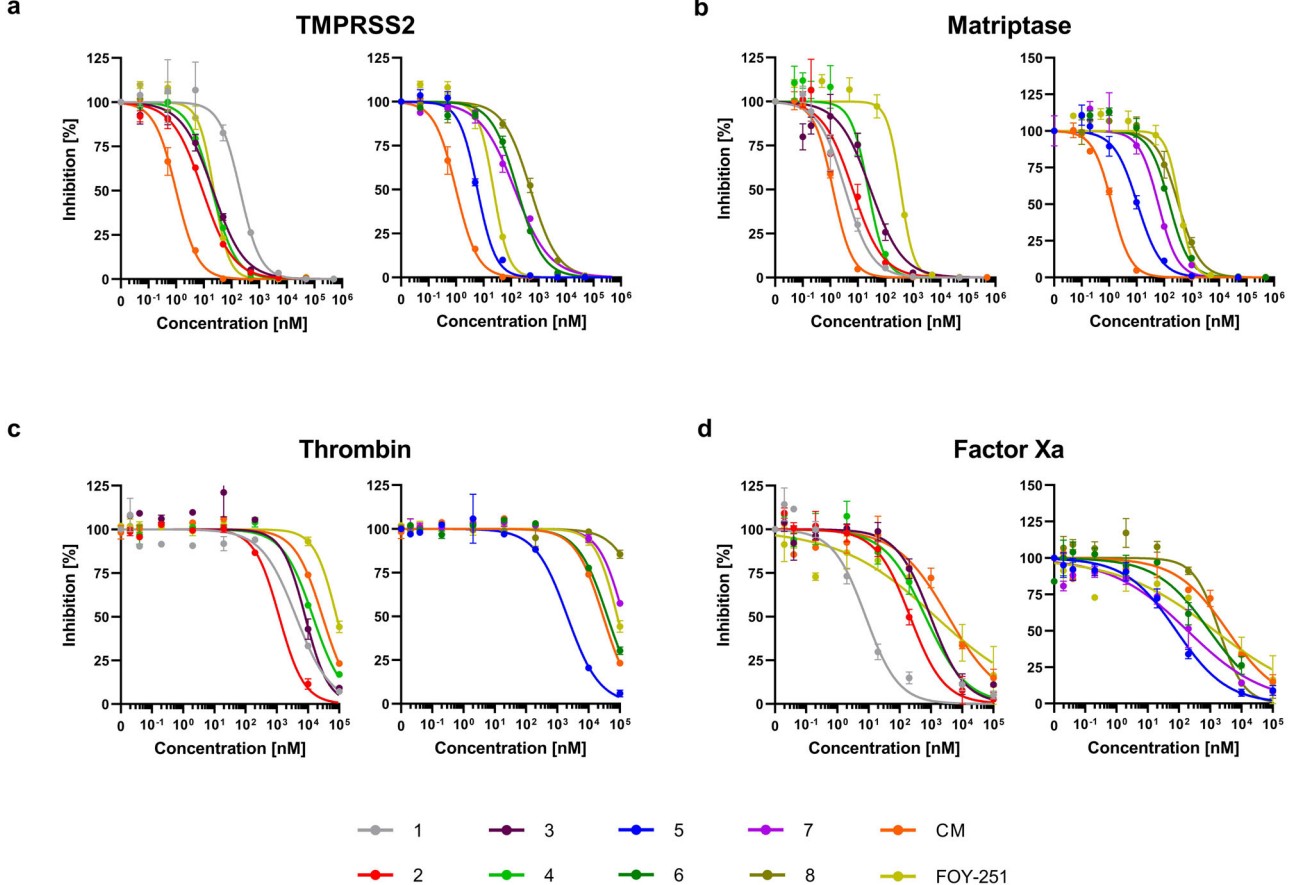

**Fig. 2 Peptidomimetic inhibitors block activity of purified proteases.** Isolated TMPRSS2 (**a**), matriptase (**b**), thrombin (**c**), and factor Xa (**d**) were mixed with peptidomimetic inhibitors, camostat mesylate (CM), and FOY-251. After 30 min, the fluorogenic reference substrate Boc-Gln-Ala-Arg-AMC was added to TMPRSS2 or matriptase and the chromogenic substrates D-Phe-Homopro-Arg-pNA or Bz-Ile-Glu-Gly-Arg-pNA were added to thrombin or factor Xa, respectively. The velocity of substrate degradation was assessed by recording the fluorescence intensity at 460 nm or the absorbance at 405 nm within 2 h. Shown are the means ± SD of $n = 1$ experiment performed in triplicates.

**Table 2 Inhibitory activity ($K_i$) of synthesized TMPRSS2 inhibitors 1–8 against TMPRSS2, matriptase, thrombin, and factor Xa.**

| | $K_i$ [nM] | | | | Selectivity indices | | |
|---|---|---|---|---|---|---|---|
| Compound | TMPRSS2 | Matriptase | Thrombin | Factor Xa | Matriptase | Thrombin | Factor Xa |
| 1 | 86.7 | 1.7 | 2077 | 4.1 | 0.02 | 24 | 0.05 |
| 2 | 3.8 | 3.3 | 599 | 106.7 | 0.9 | 158 | 28.1 |
| 3 | 9.1 | 14.0 | 4088 | 271.3 | 1.5 | 449 | 29.8 |
| 4 | 8.5 | 11.5 | 7217 | 331 | 1.4 | 849 | 38.9 |
| 5 | 2.5 | 5.2 | 1046 | 41.1 | 2.1 | 418 | 16.4 |
| 6 | 71.5 | 75.6 | >50,000 | 472.4 | 1.1 | >699 | 6.6 |
| 7 | 57.7 | 30 | >50,000 | 94.1 | 0.5 | >867 | 1.6 |
| 8 | 215.9 | 159.8 | >50,000 | 965.5 | 0.7 | 232 | 4.5 |
| CM | 0.4 | 0.6 | >50,000 | 1785.5 | 1.5 | >12,500 | 4,464 |
| FOY-251 | 9.7 | 173.4 | >50,000 | 697 | 17.9 | >5155 | 71.9 |

Selectivity indices represent the quotient of $K_i$ values of matriptase, thrombin, and factor Xa by the $K_i$ value of TMPRSS2.

compounds 2, 4, 5, and 7 were further analyzed for inhibition of cellular TMPRSS2 activity.

Having demonstrated that the designed peptidomimetics inhibit cell-free TMPRSS2 activity, we next analyzed inhibition of cell-associated protease activity. For this, we used SARS-CoV-2 permissive Caco-2 cells, which show high levels of TMPRSS2 mRNA and express TMPRSS2 on the cell surface (Supplementary Fig. 7)[39]. Cells were incubated with the respective inhibitors and treated with fluorogenic protease substrate. The most potent inhibitors against matriptase and TMPRSS2 also efficiently prevented cell-mediated proteolysis of the fluorogenic substrate with half maximum inhibitory concentrations ($IC_{50}$) of 12.7–234.2 nM, with compounds 2 ($IC_{50} = 32$ nM) and 5 ($IC_{50} = 12.7$ nM) being most active (Fig. 4a and Supplementary Table 6). To ensure that the reduction in cellular protease activity is due to inhibition of TMPRSS2, HEK293T cells transiently expressing TMPRSS2 were treated with the respective inhibitors and fluorogenic protease substrate. The signals were corrected for the protease activity of mock-transfected HEK293T cells and revealed a dose-dependent reduction of cellular TMPRSS2

activity (Fig. 4b and Supplementary Table 6), with compounds 2 ($IC_{50}$ = 3.5 nM) and 5 ($IC_{50}$ = 2.2 nM) as the most active synthesized inhibitors. Taken together, our results demonstrate that the synthesized peptidomimetic inhibitors potently reduce the activity of purified matriptase and TMPRSS2 while showing no activity against thrombin. Further, the selected most potent inhibitors display selectivity over factor Xa and reduce cellular TMPRSS2 activity.

**TMPRSS2-specific peptidomimetic inhibitors block SARS-CoV-2 infection**. We next analyzed whether compounds 1–8 may inhibit SARS-CoV-2 spike-driven viral entry. For this, Caco-2 cells treated with serial dilutions of the compounds (and CM as control) were inoculated with luciferase encoding lentiviral pseudoparticles carrying the wild-type SARS-CoV-2 spike protein. Transduction rates were determined 2 days later by measuring cell-associated luciferase activity and showed a concentration-dependent inhibition of viral entry for all analyzed compounds (Fig. 5a). Compound 5 was most efficient with an $IC_{50}$ value of 467.2 nM and was even more potent than CM ($IC_{50}$ ~ 747.5 nM). Compounds 1–4 and 7 suppressed spike-driven entry with $IC_{50}$ values between 1200 and 2068 nM while compounds 6 and 8 were the least antivirally active with $IC_{50}$

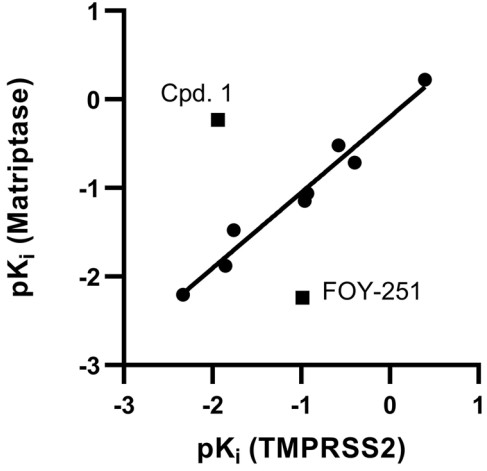

**Fig. 3 Correlation of the $pK_i$ values from compounds 1–8, CM and FOY-251 against TMPRSS2 and matriptase.** A linear regression fit ($R^2$ = 0.96) was plotted through the data points excluding cpd. 1 and FOY-251.

values between 5604–12,085 nM, respectively (Supplementary Table 7).

As the wild-type virus has largely been replaced by SARS-CoV-2 variants of concern (VOC) with increased transmissibility, virulence, or immune escape, we also determined the activity of selected compounds against the Alpha, Beta, Delta, and Omicron BA.1 spike protein. TMPRSS2 inhibitors 2, 4, 5, and 7 suppressed cell entry mediated by all spike variants in a dose-dependent manner with $IC_{50}$ values ranging between 260.7 and 2367 nM (Fig. 5b–d and Supplementary Table 7). Collectively these data show that the designed TMPRSS2 inhibitors suppress SARS-CoV-2 spike-driven viral transduction.

We next analyzed whether the inhibitors may also block authentic SARS-CoV-2 infection. To this end, Caco-2 cells were supplemented with serial dilutions of the compounds 1–8 or CM and were then infected with SARS-CoV-2 strain Wuhan-Hu-1. Infection rates were determined 2 days later by quantifying intracellular viral protein expression by ELISA[40]. All compounds including CM suppressed SARS-CoV-2 infection in a concentration-dependent manner (Fig. 6a). The $IC_{50}$ values were, however, generally higher as compared to the pseudotype experiment (Supplementary Table 8). Compounds 2, 4, and 5 were the most potent inhibitors with $IC_{50}$ values of 4.6, 5.7, and 4.7 μM, respectively, similar to CM (3.6 μM). The remaining compounds were less active with $IC_{50}$ values >10 μM. The DMSO solvent control neither affected the transduction of cells with SARS-CoV-2 pseudoparticles nor the infection with wild-type virus (Supplementary Fig. 8) and we did not observe cytotoxic effects from the compounds tested that exceeded the toxicity of the solvent control (Fig. 6e and Supplementary Fig. 9).

Finally, we determined the antiviral activity of compounds 2, 4, 5, and 7 against a SARS-CoV-2 isolate harboring the D614G mutation, which increases viral infectivity (Fig. 6b), and the VOCs Alpha (Fig. 6c) and Beta (Fig. 6d)[41]. The four selected compounds as well as CM inhibited all three tested SARS-CoV-2 isolates. Compounds 2 and 5 suppressed the infection of the SARS-CoV-2 Wuhan-Hu-1 D614G strain with $IC_{50}$ values of 17.1 and 11.9 μM, respectively, and were even more active than CM (26.7 μM) (Supplementary Table 8). SARS-CoV-2 variants Alpha and Beta were most efficiently inhibited by compound 5 ($IC_{50}$ of 6.8 and 6.3 μM, respectively) and CM ($IC_{50}$ of 16.4 and 9.3 μM, respectively). Compounds 2, 4, and 7 showed moderately higher $IC_{50}$ as compared to CM and compound 5 (Supplementary Table 8) but were still capable of blocking infection entirely. Thus, the designed peptidomimetic TMPRSS2 inhibitors prevent SARS-CoV-2 infection with comparable antiviral activity as

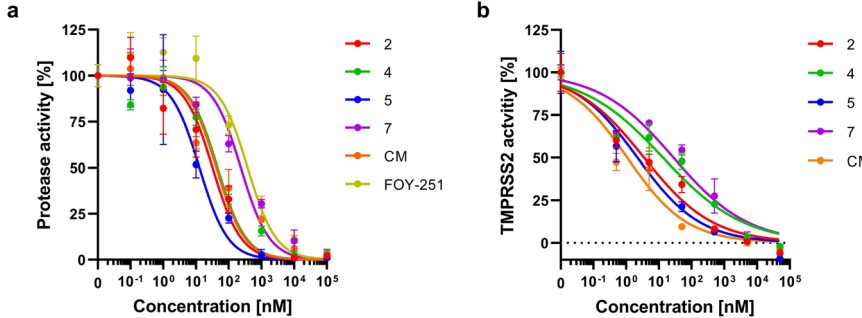

**Fig. 4 Peptidomimetic inhibitors block cellular protease activity. a** Peptidomimetic compounds 2, 4, 5, and 7 as well as camostat mesylate (CM) and FOY-251 were added to Caco-2 cells. After 30 min, the fluorogenic reference substrate Boc-Gln-Ala-Arg-AMC was added, and the reaction rate of substrate degradation was assessed by recording the fluorescence intensity within 2 h. **b** Peptidomimetic compounds 2, 4, 5, and 7 as well as camostat mesylate (CM) were added to HEK293T cells transiently expressing TMPRSS2, followed by addition of fluorogenic reference substrate Boc-Gln-Ala-Arg-AMC. Graph shows normalized fluorescence intensities after 2 h, corrected for the signal of mock-transfected HEK293T cells. Shown are the means ± SD of $n$ = 1 experiment performed in triplicates (**a**) or duplicates (**b**). Calculated $IC_{50}$ values for each compound are presented in Supplementary Table 6.

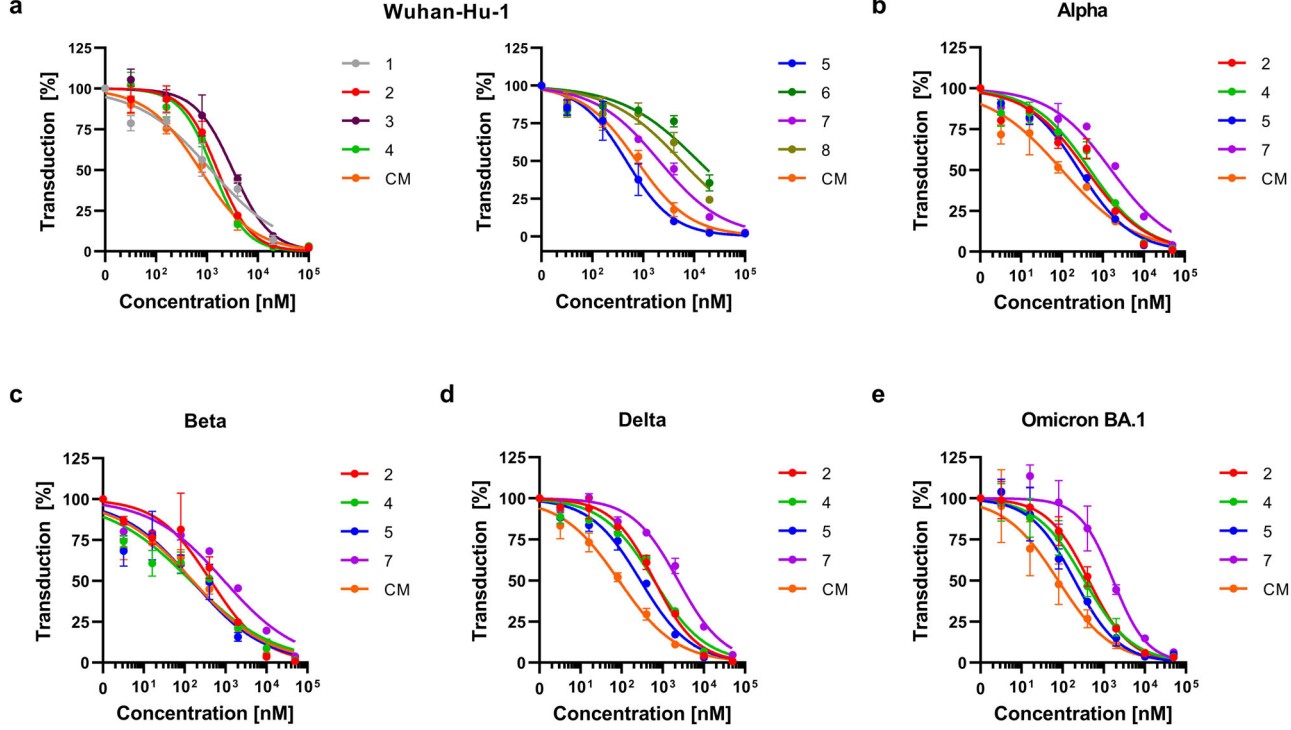

**Fig. 5 Peptidomimetic inhibitors reduce SARS-CoV-2 spike-driven entry.** Peptidomimetic inhibitors and the small molecule camostat mesylate (CM) were added to Caco-2 cells. After 1 h, cells were transduced with lentiviral SARS-CoV-2 pseudoparticles carrying the spike protein of SARS-CoV-2 wild-type (**a**), Alpha (**b**), Beta (**c**), Delta (**d**), or Omicron BA.1 (**e**) variant of concern. Transduction rates were assessed 2 days post transduction by measuring luciferase activity in cell lysates. Shown are the means ± SEM of $n = 2$ independent experiments, each performed in triplicates. Calculated $IC_{50}$ values for each compound are presented in Supplementary Table 7.

CM, which is currently evaluated in clinical trials as COVID-19 therapeutic.

**In vitro stability of selected leads in body fluids and epimerization studies.** The stability of peptidomimetic inhibitors is the main challenge in peptide drug development. Peptidases in blood might degrade the inhibitor before reaching the desired target in an adequate time. Therefore, the stability of two compounds in body fluids was assessed. Compound **2** was selected due to the high selectivity as well as potency in suppressing genuine SARS-CoV-2 infection, and compound **7** was selected since the structure comprises non-proteinogenic amino acids with potential resistance to proteolysis.

In the first experiment, compound 7 was spiked into 25% human serum and chromatographic analysis of peptide content confirmed the presence of residual inhibitor for up to 10 days (Supplementary Fig. 10). Due to the strong electrophilicity of the ketobenzothiazole serine trap[42], the P1 Arg $C_\alpha$-atom of compound 7 atom readily epimerized within 30 min (Supplementary Fig. 10). Of note, both epimers displayed similar inhibitory activity, as shown for compound 2 (Supplementary Fig. 11). The assay buffer used for in vitro activity studies did not alter the epimerization ratio significantly (Supplementary Fig. 12). We then determined the residual TMPRSS2 inhibitory activity of compounds 2 and 7 and camostat mesylate after incubation in human serum (Fig. 7a–c). Compounds 2 and 7 displayed similar activity than in assay buffer and both inhibitors retained their activity against TMPRSS2 for all timepoints tested, with an increase in $K_i$ values of 1.4–1.8-fold during the course of 10 days (Fig. 7d). In contrast, the activity of the control inhibitor camostat mesylate reduced by around 60-fold upon addition to serum. Further reduction in activity was observed up to day 1, while the

inhibition vanished entirely within 10 days in human serum. Similar observations were made for incubation of the compounds in human plasma and serum-free cell culture medium (Supplementary Figs. 13 and 14). Therefore, the stability of the peptidomimetic inhibitors supports further studies in in vivo models.

## Discussion

We here describe potent and stable peptidomimetic inhibitors of TMPRSS2 that block SARS-CoV-2 infection. Targeting TMPRSS2 is a promising antiviral strategy because the protease is not only essential for SARS-CoV-2 entry, but also primes glycoproteins of various other viruses for subsequent fusion and infection. Since TMPRSS2 is a host and not a viral protein, TMPRSS2-targeting therapeutics that block its enzymatic activity should be less likely to induce resistance mutations.

To develop TMPRSS2 inhibitors, we used literature data on TMPRSS2 substrate preferences and designed a reference binder, which was used as a template for the preparation of peptidomimetic libraries, which then were screened in silico against the binding cavity of matriptase as a surrogate model for TMPRSS2 and against a TMPRSS2 homology model. A library of recognition sequences was compiled by incorporation of the identified top-scoring amino acids in the template. The recognition sequences were connected with an electrophilic ketobenzothiazole serine trap moiety as a reactive functional group to yield a panel of inhibitors[43–45]. We tested the inhibitors against isolated enzymes and our data identified the four compounds 2, 4, 5, and 7 as potential hits with high activity against TMPRSS2 and matriptase, and good off-target selectivity against coagulation proteins thrombin and factor Xa. CM and its rapidly forming active metabolite FOY-251 were used for comparison since they

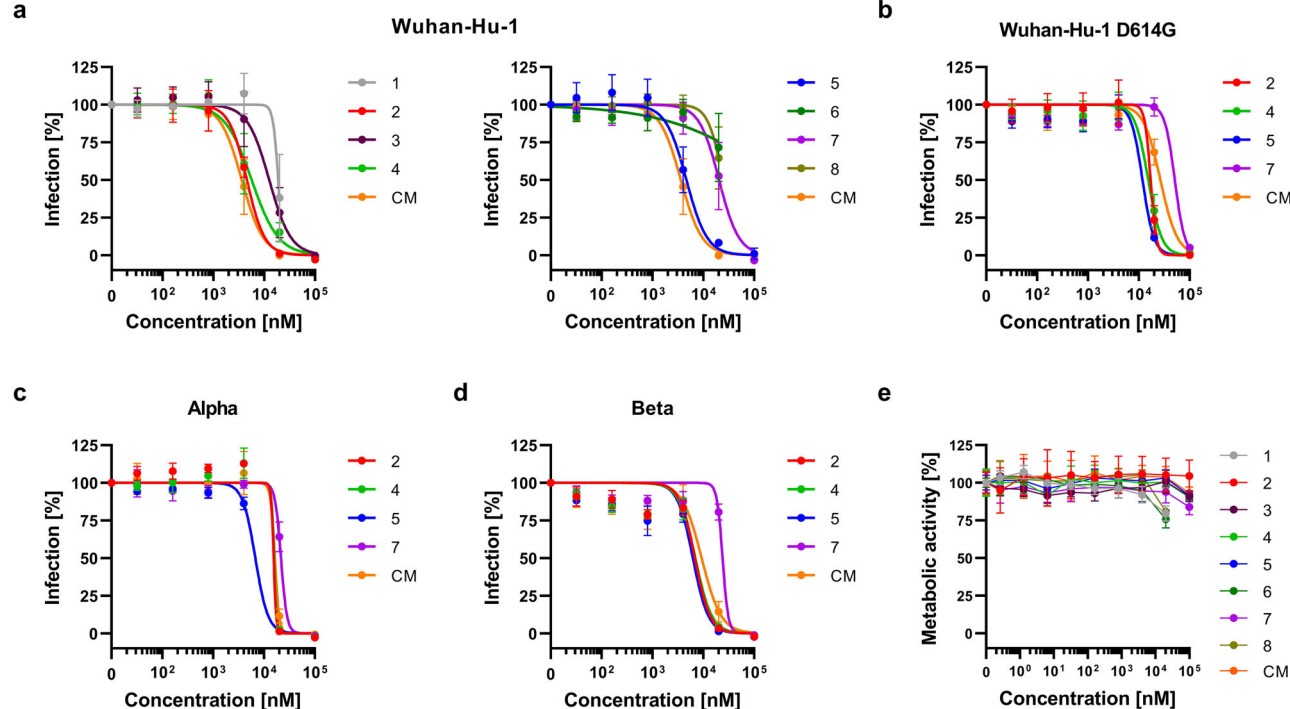

**Fig. 6 Peptidomimetic inhibitors reduce SARS-CoV-2 infection.** Peptidomimetic inhibitors and the small molecule camostat mesylate (CM) were added to Caco-2 cells. After 1 h, cells were infected with SARS-CoV-2 Wuhan-Hu-1 (**a**), SARS-CoV-2 bearing the spike D614G mutation (**b**), or the variants of concern Alpha (**c**) and Beta (**d**). Infection rates were determined 2 days post infection by in cell ELISA for the viral N protein. (**e**) Cytotoxicity of peptidomimetic inhibitors. Inhibitors and the small molecule camostat mesylate (CM) were added to Caco-2 cells. Cell viability was assessed 2 days post addition by measuring ATP content in cell lysates. Due to low stock concentration compounds 1, 6, and 8 were tested at a maximum concentration of 20,000 nM. Shown are the means ± SEM of $n = 3$ independent experiments (**a–d**) or mean ± SD of $n = 1$ experiment (**e**), each performed in triplicates. Calculated $IC_{50}$ values for each compound are listed in Supplementary Table 8.

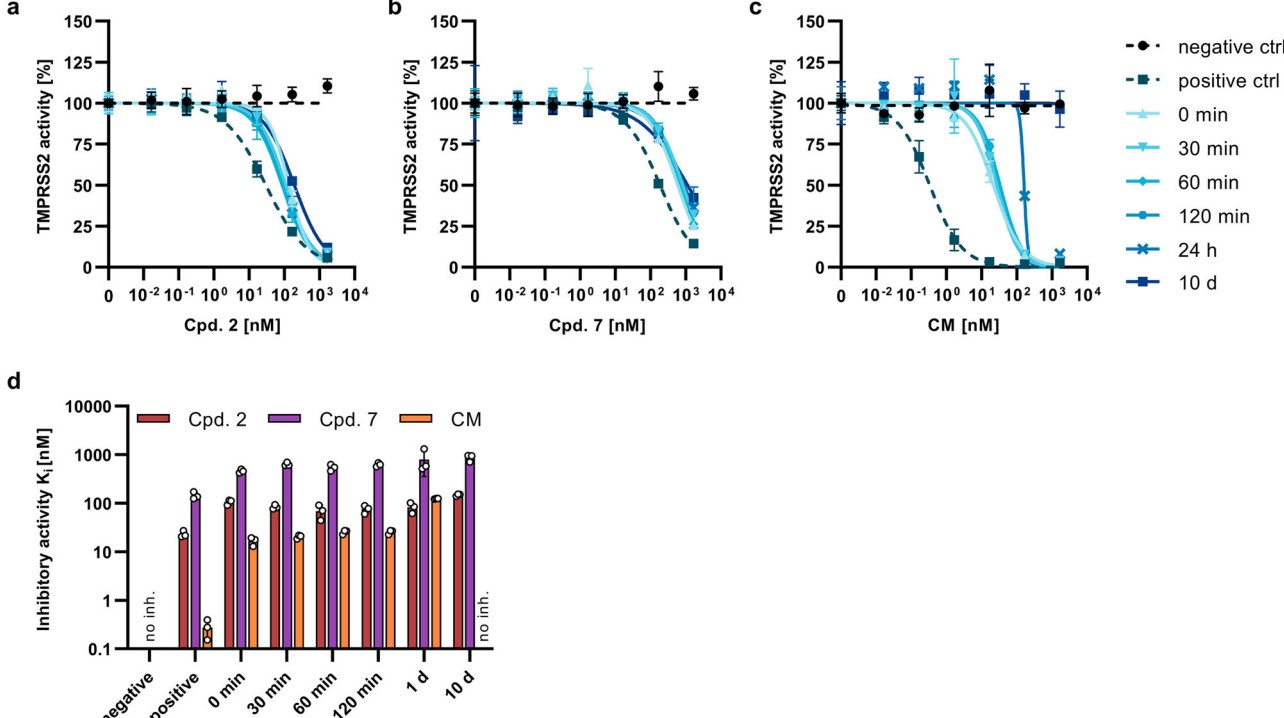

**Fig. 7 Serum stability of inhibitors.** Compound 2 (**a**), compound 7 (**b**), or camostat mesylate (**c**) were incubated in human serum for indicated timepoints. Samples were mixed with recombinant TMPRSS2, followed by the addition of the fluorogenic reference substrate BOC-Gln-Ala-Arg-AMC. Graph shows normalized fluorescence intensities after incubation for 2 h. Negative ctrl: no inhibitor, positive ctrl: inhibitor in assay buffer. **d** Inhibitory constants $K_i$ as determined from (**a–c**). Shown are the means ± SD of $n = 1$ experiment performed in triplicates. No inh. no inhibition.

have been shown previously to efficiently inhibit TMPRSS2 proteolytic activity and CM is currently evaluated in clinical trials for COVID-19[19,46]. Our best candidates show inhibitory activities in the same range as CM, and the compounds 2 and 5 even show a 2–3-fold higher activity against isolated TMPRSS2 than FOY-251. The ketobenzothiazole serine trap moiety revealed the highest activity which may be attributed to a preferential fit in the hydrophobic S1' pocket of TMPRSS2, in contrast to the less hydrophobic and smaller ketothiazole. The addition of further amino acids or mimetics on the ketobenzothiazole moiety could improve the interaction with the S'-sites and might be of interest in further studies[43].

Furthermore, the high activity of the top compounds was confirmed through cleavage of a fluorogenic substrate on Caco-2 epithelial cells, which serve as an intestinal model carrying the TMPRSS2 protease on the surface, and TMPRSS2 expressing HEK293T cells. Finally, the inhibitors blocked SARS-CoV-2 spike-driven viral entry into and infection of Caco-2 cells by authentic SARS-CoV-2 wild-type and variants of concern in a concentration-dependent manner. Thus, the designed inhibitors likely block TMPRSS2 mediated proteolytic priming of the viral spike protein, thereby preventing subsequent receptor binding and fusion. These data also show that the tested SARS-CoV-2 VOCs are still dependent on TMPRSS2 as essential cofactor for cell entry and demonstrate that VOCs that escape from pre-existing immunity are equally sensitive to entry inhibitors, as previously shown for soluble ACE2 or fusion-inhibiting peptide EK1 and EK1C4[11,47].

Two compounds (2 and 7) were incubated in body fluids for up to 10 days and retained inhibitory activity in the sub-nanomolar range which is remarkable considering the literature known stability issues of peptide therapeutics[48,49]. Accordingly, CM was found inactive when incubated for the same period of time. The rapid epimerization of the compounds in blood serum did not alter the activity significantly, suggesting that structurally simplified inhibitors may be developed. The high stability in body fluids and potent anti-TMPRSS2 and anti-SARS-CoV-2 activity warrants further preclinical development of selected compounds. Furthermore, a combination with drugs targeting viral replication, such as the protease inhibitor nirmatrelvir (in combination with ritonavir, Paxlovid®), could yield synergistic effects. Of note, the TMPRSS2 inhibitors will not only act against SARS-CoV-2 but potentially also block other TMPRSS2-dependent coronaviruses such as SARS-CoV and MERS-CoV, and likely also future novel emerging coronaviruses, and TMPRSS2-dependent viruses from other viral families. In sum, TMPRSS2 represents an attractive drug target in COVID-19, and downregulation of its enzymatic activity with active and selective inhibitors should significantly improve health rehabilitation. Here we showed a new direction for the fast development of peptidomimetic inhibitors and our results offer potential candidates with comparable activities to CM whose efficacy may be further elucidated in in vivo studies.

## Methods

**Molecular modeling**. When modeling was performed, no crystal structure of TMPRSS2 was freely available in the protein data bank (PDB)[37]. Thus, a TMPRSS2 homology model was built using the Swiss-Model web server[50]. The template structure was selected based on the serine-protease hepsin in complex with N-acetyl-6-ammonio-L-norleucyl-L-glutaminyl-N-[(1 S)-4-w-1-(chloroacetyl)butyl]-L-leucinamide (PDB-ID: 1Z8G)[51] with a sequence identity to TMPRSS2 of 42.49% and a 1.5 Å resolution. For subsequent docking studies, both the homology model and the crystal structure of matriptase in complex with N-(3-phenylpropanoyl)-3-(1,3-thiazol-4-yl)-L-alanyl-N-[(1 S,2 S)-1-(1,3-benzothiazol-2-yl)-5-carbamimidamido-1-hydroxypentan-2-yl]-L-valinamide (PDB-ID: 6N4T)[36] as a surrogate model were used. The focused serine-protease inhibitor library was derived from the ZINC15 database[52]. Sequences of tripeptides for docking studies on TMPRSS2 homology/surrogate were generated using CycloPs and included proteinogenic and

non-proteinogenic amino acids (aa)[53]. The generated SMILES were modified to carry a N-terminal acetyl cap (ace) and a C-terminal aldehyde serine trap. Prior to docking, all molecules were protonated and energetically minimized using MOE2019[54]. Hereby, the MMFF94x[55] forcefield was used for small molecules and AMBER14:EHT[56] for peptidic molecules. For molecular docking with LeadIT-2.3.2, the binding site was defined to include all residues within 6 Å around the reference ligand of the hepsin homology model (PDB-ID: 1Z8G)[57]. For matriptase, all residues within 6.5 Å around the crystallographic reference ligand (PDB-ID: 6N4T) and water molecules forming at least three interactions with the target and ligand were included. Structures were protonated with the Protoss module[58] within LeadIT-2.3.2. All dockings were performed using standard settings and the enthalpy–entropy hybrid approach. The docking strategy was validated for matriptase surrogate model by redocking of the ligand (PDB-ID: 6N4T), and for TMPRSS2 homology model and matriptase surrogate model by docking of the substrate ace-D-Arg-Pro/Gly-Arg-nme, and by a binder vs. non-binder discrimination using 56 published TMPRSS2 inhibitors and 314 decoys generated for four inhibitors present at ZINC using the database of useful decoys enhanced (DUD-E)[31,32,59]. Results were analyzed by FlexX score and visual pose inspection to select molecules for purchase and synthesis. During the course of this study the crystal structure of TMPRSS2 in a covalent complex with nafamostat became available (PDB-ID: 7MEQ)[38]. Hence, retrospective docking studies were performed with this structure as well. After untethering the covalent bond, the binding site was defined to include residues 8 Å around the reference ligand and additionally residues Leu-419, Lys-340, Thr-341 from the S2-S4 sites and water molecules forming at least three interactions with the protease (water molecules 703,735 and 792 within the S1 pocket) were included. The LeadIT docking parameters were as described for the hepsin-based homology model and the surrogate matriptase. The docking setup was validated by redocking of 4-guanidinobenzoic acid (from nafamostat) and binder-decoy discrimination. Figures were made with PyMOL[60].

**Chlorination of tritylhydroxide resin (Trt-OH)**. The chlorination of tritylhydr-oxide resin was performed based on a modified method described elsewhere[61]. In short, a 250-mL round-bottom flask was rinsed with dry dichloromethane (DCM) and 25 g of Trt-OH (0.8 mmol/g, mesh 100–200, company: Iris Biotech) was added. The resin was suspended in a mixture of 50% DCM and 50% toluene, just enough to double the resin volume followed by 10 mL acetyl chloride. The glass vial was sealed and agitated for 24 h. The next day, the resin was dried and thoroughly washed with DCM (4 × 5 mL). Chlorinated resin (Trt-Cl) was stored in a freezer.

**Addition of first amino acid to Trt-Cl**. In all, 15 mL of dry DCM was added to 0.8 g of Trt-Cl[61] resin (0.8 mmol/g) and shaken for 10 min. Next, 2.5 mmol (2.5 mmol) of amino acid and 5 equiv. (5 mmol) of activator base were added and the mixture was shaken overnight at room temperature. The next day, the resin was filtered and washed with N,N-dimethylformamide (DMF) (2 × 10 mL) and DCM (2 × 10 mL). Prior to use for solid-phase peptide synthesis, the resin was swollen with 10 mL DMF.

**Solid-phase peptide synthesis**. Peptides were synthesized as C-terminal amides on a loaded 2-chlorotrityl chloride resin using 9-fluorenylmethoxycarbonyl (Fmoc) strategy. All Fmoc-protected amino acids were purchased from Novabiochem, as well as benzotriazol-1-yloxytripyrrolidinophosphonium hexafluorophosphate (PyBOP), N,N-diisopropylethylamine (DIEA), DMF, DCM and were used as received. All Fmoc-protected amino acids were dissolved in DMF at a concentration of 0.2 M. The coupling was done with the standard procedure of solid-phase peptide synthesis[62]. In short, Fmoc deprotection was performed by treating the peptidyl resin with 20% piperidine in DMF for 10 min at 70 °C. After the reaction, the resin was washed with DMF and DCM, each (2 × 10 mL), then filtrated. The coupling was done by addition of 7.5 mL of Fmoc-protected amino acid (3 equiv.), 3 mL of 1 M PyBOP (3 equiv.) and 1.5 mL of 0.5 M DIEA (3 equiv.). The reaction mixture was shaken at 75 °C for 30 min. Afterward, the resin was washed as described above and used for the next step.

**N-Cap modification and cleavage from the resin**. The N-terminus of the peptides with protected side chains were acetyl capped using 20 mL of 0.5 M Ac₂O/DMF and 1 M DIEA/DMF. The reaction was shaken for 2 h at room temperature. The resin was filtered and washed with DMF and DCM, each (2 × 10 mL). The cleavage of the peptide with protected side chains was accomplished using 20 mL of 20% hexafluoroisopropanol (HFIP) in DCM. The mixture was put on an orbital shaker for 3 h at room temperature and then filtered. The solvent was taken off in vacuo and the peptide was precipitated using 50 mL of diethyl ether.

**Synthesis of Boc-Arg(Mtr) weinreb amide (S1)**. ((S)-tert-Butyl-(1-(methox-y(methyl)amino)-5-(3-((4-methoxy-2,3,6-trimethylphenyl)sulfonyl)guanidino)-1-oxopentan-2-yl)carbamate)[63]. To a solution of N-(tert-butoxycarbonyl)-N-((4-meth-oxy-2,3,6-trimethylphenyl)sulfonyl)-L-arginine (1.56 g, 3.21 mmol, 1.0 equiv.) in 34 mL THF, N,O-dimethylhydroxyl-amine hydrochloride (0.64 g, 6.42 mmol, 2.0 equiv) and 1-hydroxybenzotriazole hydrate (0.54 g, 3.53 mmol, 1.1 equiv.) were dis-solved at room temperature. DIEA (1.63 mL, 9.63 mmol, 3 equiv.) and EDCl (0.65 g, 3.37 mmol, 1.05 equiv.) were added and the solution was stirred for 4.5 h. The mixture

was concentrated in vacuo and the reaction mixture extracted with ethyl acetate (150 mL), washed with 5% aqueous acetic acid (75 mL), sat. aqueous NaHCO$_3$ (75 mL), water (75 mL), and brine (75 mL). The organic layer was dried over MgSO$_4$, filtered and concentrated in vacuo to yield the Weinreb amide S1 (0.86 g, 1.6 mmol, yield: 69%) as a white powder (Supplementary Fig. 1). Purity (HPLC, 220 nm) > 80% [1].H NMR (300 MHz, CD3OD): δ = 1.43 (s, 9 H) 1.49–1.60 (m, 4 H) 1.87 (s, 1 H) 2.13 (s, 3 H) 2.61 (s, 3 H) 2.67 (s, 3 H) 3.17 (m, 3 H) 3.74 (s, 3 H) 3.90 (s, 3 H) 6.67 ppm (s, 1 H). MS (ESI): $m/z$: calcd. for C$_{23}$H$_{39}$N$_5$O$_7$S$_2$ [M + H]$^+$ 530.3, [2 M + H]$^+$ 1059.6, found [M + H]$^+$ 530.2, [2 M + H]$^+$ 1059.4.

**Synthesis of Boc-Arg(Mtr) ketobenzothiazole (S2)**. ((S)-*tert*-Butyl-(1-(benzo[*d*]thiazol-2-yl)-5-(3-((4-methoxy-2,3,6-trimethylphenyl)sulfonyl)guanidino)-1-oxo-pentan-2-yl)carbamate)[64]. To a solution of benzothiazole (0.47 g, 3.53 mmol, 1.1 equiv.), *n*-BuLi (1.6 M, 4 mL, 6.42 mmol, 2 equiv.) was added dropwise to THF (50 mL). After the mixture was stirred for an additional 30 min, Boc-Arg(Mtr) Weinreb amide (1.94 g, 3.21 mmol, 1 equiv.) was dissolved in THF (15 mL) and added slowly over 50 min. The mixture was stirred at –78.8 °C for 3 h. The reaction was quenched with sat. aqueous NH$_4$Cl (30 mL) and the aqueous layer was extracted with EtOAc (40 mL). The organic phase was collected, dried with Na$_2$SO$_4$, and then concentrated. The resulting residue was purified by semi-preparative RP-HPLC to yield the compound S2 (0.32 g, 0.5 mmol, yield: 35%) as a yellow powder (Supplementary Fig. 1). Purity (HPLC, 220 nm) >95%. $^1$H NMR (300 MHz, CD3OD): δ = 1.43 (s, 9 H) 1.66 (m, 4 H) 2.05 (s, 3 H) 2.55 (s, 3 H) 2.70 (s, 3 H) 3.23 (m, 2 H) 3.81 (s, 3 H) 5.32 (m, 1 H) 6.55 (s, 1 H) 7.69 (m, 2 H) 8.27 ppm (m, 2 H). MS (ESI): $m/z$: calcd. for C$_{28}$H$_{37}$N$_5$O$_6$S$_2$ [M + H]$^+$ 604.2, [2 M + H]$^+$ 1207.4, found [M + H]$^+$ 603.9, [2 M + H]$^+$ 1206.6.

**Synthesis of HCl•H-Arg(Mtr) ketobenzothiazole (S3)**. (*N*-(*N*-(4-Amino-5-(benzo[*d*]thiazol-2-yl-5-oxopentyl)carbamimidoyl)-4-methoxy-2,3,6-trimethyl-benzenesulfonamide)[43]. Compound S2 (0.200 g, 0.40 mmol) was stirred in 1.5 M HCl/dioxane (10 mL) at room temperature for 18 h. The solvent was removed the resulting residue was dried in vacuo and purified by RP-HPLC (Supplementary Fig. 1). The received compound was used for the following peptide couplings.

**Synthesis of Boc-Arg(Mtr) ketothiazole (S4)**. ((S)-*tert*-Butyl-(5-(3-((4-methoxy-2,3,6-trimethylphenyl)sulfonyl)guanidino)-1-oxo-1-(thiazol-2-yl)pentan-2-yl)carbamate)[65]. To a solution of 2-bromothiazol (0.211 g, 1.29 mmol, 3.3 equiv.) in dry THF (10 mL) *n*-BuLi (2.5 M, 0.52 mL, 1.29 mmol, 3.3 equiv.) was added dropwise under inert atmosphere at –78 °C. The reaction mixture stirred for 1.5 h at –78 °C, followed by dropwise addition of compound S1 (0.205 g, 0.39 mmol, 1 equiv.) at the same temperature. The resulting solution was stirred 2 h at –78 °C, after which sat. aqueous NH$_4$Cl (10 mL) was added. The organic phase was separated and the aqueous phase was extracted three times with EtOAc. The combined organic extracts were washed with brine (30 mL), dried over Na$_2$SO$_4$, filtered and concentrated in vacuo. The residue was purified on a silica column eluting with EtOAc/cyclohexane (4:1 v/v), to afford the compound S4 (0.12 g, 0.22 mmol, yield: 56%) as a white foam (Supplementary Fig. 1). Purity (LC, 254 nm) > 95%.H NMR (300 MHz, CDCl$_3$): δ = 8.04 (d, 1 H), 7.72 (d, 1 H), 6.53 (s, 1 H), 5.64 (d, 1 H), 5.41 (s, 1 H), 3.83 (s, 3 H), 3.26 (m, 2 H), 2.67 (s, 3 H), 2.59 (s, 3 H), 2.12 (s, 3 H), 1.76 – 1.57 (m, 4 H), 1.41 (s, 9 H). ppm. MS (ESI): $m/z$: calcd. for C$_{24}$H$_{35}$N$_5$O$_6$S$_2$ [M + H]$^+$ 554.2, found [M + H]$^+$ 554.2.

**Synthesis of TFA•H-Arg(Mtr) ketothiazole (S5)**. ((S)-*N*-(*N*-(4-Amino-5-oxo-5-(thiazol-2-yl)pentyl)carbamimidoyl)-4-methoxy-2,3,6-trimethylbenzenesulfonamide). Compound S4 (0.256 g, 0.46 mmol) was stirred in DCM (3 mL) at 0 °C and TFA (1 mL) was added. The reaction mixture stirred for 1 h at ambient temperature, then isopropyl alcohol (0.5 mL) was added. The solution was concentrated in vacuo and triturated with diethyl ether. The supernatant was decanted and the residue was purified by RP-HPLC (Supplementary Fig. 1). The obtained compound S5 was used for the following peptide couplings.

**Synthesis of Boc-Arg(Mtr) alcohole (S6)**. ((S)-*tert*-Butyl-(1-hydroxy-5-(3-((4-methoxy-2,3,6-trimethylphenyl)sulfonyl)guanidino)pentan-2-yl)carbamate). To a solution of Boc-Arg(Mtr)-OH (0.3 g, 0.62 mmol,1 equiv.) in dry THF (5 mL) were added NMM (0.063 g, 0.62 mmol, 1 equiv.) and EtOCOCl (0.067 g, 0.62 mmol, 1 equiv.) at –15 °C under argon. The reaction mixture stirred for 1 h at –15 °C, then transferred dropwise via canula into a stirred solution of NaBH$_4$ (0.047 g, 1.24 mmol, 2 equiv.) in water (15 mL). The resulting solution was stirred 5 min at 0 °C and then diluted with water (15 mL). The aqueous phase was extracted twice with EtOAc (10 mL). The combined organic extracts were dried over Na$_2$SO$_4$ and concentrated in vacuo to obtain the compound S6 (0.24 g, 0.5 mmol, yield: 81%) as a colorless oil (Supplementary Fig. 1). Purity (LC, 254 nm) 98%[1].H NMR (300 MHz, CDCl$_3$) δ = 6.52 (s, 1 H), 6.33 (s, 2 H) 5.15 (d, 1 H), 3.82 (s, 3 H), 3.55 (s, 2 H), 3.21 (s, 1 H), 2.69–2.66 (m, 5 H), 2.59 (s, 3 H), 2.12 (s, 3 H), 1.55 (s, 4 H), 1.40 (s, 9 H) ppm.LC-MS: $m/z$: calcd. for C$_{21}$H$_{36}$N$_4$O$_6$S [M + H]$^+$ 473.2, found [M + H]$^+$ 473.2.

**Synthesis of TFA•H-Arg(Mtr) alcohol (S7)**. ((S)-*N*-(*N*-(4-Amino-5-hydroxypentyl)carbamimidoyl)-4-methoxy-2,3,6-trimethylbenzenesulfonamide).

Compound S6 (0.22 g, 0.47 mmol) was stirred in DCM (3 mL) at 0 °C and TFA (1 mL) was added. The reaction mixture stirred for 2 h at ambient temperature, then isopropyl alcohol (0.5 mL) was added. The solution was concentrated in vacuo and triturated with diethyl ether. The supernatant was decanted and the residue was purified by RP-HPLC. The received compound S7 was used for the following peptide couplings (Supplementary Fig. 1).

**Preparation of inhibitors**. Respective serine traps (1.5 equiv.) were coupled with dipeptides (1.0 equiv.) bearing standard protection groups using PyBOP (1.5 equiv.) and DIEA (3 equiv.) in DMF. After the reaction was agitated for 4 h at room temperature, 3 mL of a deprotection solution was added (93% TFA, 3.5% TIPS, 3.5% H$_2$O) and further agitated for 8 h at room temperature. After concentrating, the crude inhibitor was precipitated in 50 mL cold diethyl ether and afterward purified using RP-HPLC.

**Purification, lyophilization, and analysis of peptide- and serine trap precursors and inhibitors**. All precursor compounds were purified with a semi-preparative RP-HPLC. The following gradient was applied: 95% H$_2$O/5% ACN to 5% H$_2$O/95% ACN in 30 min. Trifluoracetic acid for deprotection was dissolved in the water to a concentration of 0.1%. Column used: Zorbax Eclipse XDB C-18 9.4 × 250 mm 5 µm, company: Agilent Technologies. Detector: UV Vis detector model S-3702, company: Soma. For the detection, a wavelength of 220 nm was used. After chromatographic purification, the fractions were collected and freeze-dried overnight. The purified and lyophilized precursor compounds were stored in the freezer at −20 °C. The mass of the purified compounds was determined with MS-ESI. Model used: expression-L compact mass spectrometer, company Advion. Peptides were dissolved to a concentration of $c = 0.01$ mg/mL in MeOH + 0.1% formic acid. Injection was done by a syringe pump with a flow rate of 10 µL/min.

**Enzymes**. Recombinant human TMPRSS2 was purchased from Creative BioMart (New York, USA) or LSBio (Seattle, USA), factor Xa was obtained from Bio-Techne GmbH (Wiesbaden, Germany). Recombinant human thrombin and matriptase protein were purchased from R&D Systems (Minneapolis, MN, USA).

**Determination of inhibitory constant $K_i$**. The activity of the compounds against the recombinant human enzymes was determined in enzyme inhibition assays. Here, a ten-point dilution series for the inhibitors was prepared and incubated for 30 min with the enzyme in TNC buffer (25 mM Tris, 150 mM NaCl, 5 mM CaCl$_2$, 0.01% Triton X-100, pH = 8) prior to adding a fluorogenic reference substrate Boc-Gln-Ala-Arg-AMC for matriptase and TMPRSS2 or a chromogenic substrate D-Phe-Homo-pro-Arg-pNA for thrombin and factor Xa, respectively. The measurements were performed on a Tecan infinite® M1000 and the fluorescence intensity was measured by exciting the AMC fluorophore at 380 nm wavelength and recording emission at 460 nm wavelength. The absorption of pNA was measured at 405 nm. Fluorescence intensities and absorption were measured every 2 min for 2 h or as endpoint after 2 h. The end concentrations of the enzymes were 0.2 nM (matriptase) and 0.2 nM (TMPRSS2) in 20 µL total volume and 0.6 nM (thrombin), 0.35 nM (factor Xa) in 100 µL total volume. The end concentration of the reference substrate was 100 µM (matriptase), 100 µM (TMPRSS2), 200 µM (factor Xa) and 100 µM (thrombin). To determine the IC$_{50}$ values, the concentration-response data were plotted with the program GraphPad prism version 8.4.2 (San Diego, California) and a nonlinear regression fit with the equation [Inhibitor] vs. normalized response was applied. The inhibitory constant $K_i$ was calculated from the IC$_{50}$ values using the Cheng–Prusoff equation ($K_i = IC_{50}/[S]/K_M$) for competitive reversible inhibitors[66]. The $K_M$ value was determined to be 77 µM for TMPRSS2 (Supplementary Fig. 2).

**Analysis of cellular TMPRSS2 expression**. The human colorectal adenocarcinoma cell line Caco-2 from the Collection of Microorganisms and Cell Cultures (DSMZ, Germany) was maintained in Eagle's Minimum Essential Medium (EMEM) supplemented with 10% FBS, 100 U /mL penicillin, 100 mg/mL streptomycin, and 2 mM glutamine (all Invitrogen, Germany). For the validation of the expression of the transmembrane serine-protease TMPRSS2, 100,000 Caco-2 cells were resuspended in 100 µL of Dulbecco's phosphate-buffered saline (DPBS, Sigma-Aldrich) and incubated with the TMPRSS2 antibody (ThermoFisher Scientific, PA5-14264) at final concentrations of 10, 20, 40, and 100 µg/mL for 30 min at 4 °C. After the separation from unbound antibody molecules by centrifugation (200 × $g$ for 3 min) and resuspension of the cells in 100 µL DPBS, 1 µL of a FITC-labeled secondary donkey anti-rabbit IgG (ThermoFisher, A16024) was added and incubated for 30 min at 4 °C. Following a final centrifugation (200 × $g$ for 3 min), the cells were resuspended in 1 mL DPBS and analyzed by flow cytometry. The measurements were performed on an Attune™ NxT cytometer (ThermoFisher) with a 488 nm laser for excitation of bound secondary antibody molecules (FITC) and a 530/30 nm band pass filter for emission detection. Using the Attune™ NxT software (ThermoFisher), Caco-2 cells were selected by the FSC/SSC plot, thereby excluding cell debris. From this dot plot gating of Caco-2 cells, a histogram plot of the BL1-H emission filter signal was generated. The signal of untreated Caco-2 cells (autofluorescence) was gated to one percent, whereby all other samples refer to the percentage of events within this gate. For the data analysis, GraphPad Prism version 8.4.2 was applied.

**Analysis of inhibition of cellular TMPRSS2 activity**. In total, 10,000 Caco-2 cells (ATCC) in 100 μL EMEM medium supplemented with 10% FBS, 100 U/mL penicillin, 100 mg/mL streptomycin, and 2 mM glutamine (all Invitrogen, Germany) were seeded per well in a 96-well plate and incubated at 37 °C for 4 days until full confluency of the cells. The cells were washed two times with PBS and EMEM medium without FBS was added. For the determination of $IC_{50}$ values, 1 μL of inhibitor was incubated for 30 min at room temperature prior to adding 2 μL of 10 mM reference substrate (Boc-Gln-Ala-Arg-AMC). The fluorescence intensity was measured as described above. In all, 20,000 HEK293T cells (ATCC) were seeded in 100 μL DMEM supplemented with 10% FBS, 100 U/mL penicillin, 100 mg/mL streptomycin, and 2 mM glutamine. The next day, cells were transfected with 100 ng TMPRSS2 expression plasmid (Addgene 53887, kindly provided by Roger Reeves, Johns Hopkins University, Baltimore, USA) using polyethyleneimine (PEI). Briefly, DNA was mixed with PEI at a DNA:PEI ratio of 1:3 in serum-free medium, incubated for 20 min at room temperature, and added to the cells. After 14 h, transfection mix was removed from the cells and 80 μL fresh medium without FBS and 10 μL inhibitors were added (final concentration 50 μM). After 15 min, 20 μL of fluorogenic reference substrate Boc-Gln-Ala-Arg-AMC were added (final concentration 100 μM). Fluorescence intensity was measured after 2 h at 37 °C as described above.

**SARS-CoV-2 pseudoparticles**. To generate replication-deficient lentiviral pseudoparticles carrying the SARS-CoV-2 spike protein (LV(Luc)-CoV-2), 900,000 HEK293T cells were seeded in 2 mL DMEM supplemented with 10% FBS, 100 U/mL penicillin, 100 mg/mL streptomycin, and 2 mM glutamine. The next day, the medium was refreshed and cells were transfected with a total of 1 μg DNA using polyethyleneimine (PEI). To this end, 2% of SARS-2 spike plasmid (encoding the spike protein of SARS-CoV-2 isolate Wuhan-Hu-1, NCBI reference sequence YP_009724390.1, SARS-CoV-2 variant Alpha (B.1.1.7), Beta (B.1.351), Delta (B.1.617.2) or Omicron BA.1 (B.1.1.529) were mixed with pCMVdR8_91 (encoding HIV structural proteins gag and pol) and pSEW-Luc2 (crippled lentiviral vector encoding the luciferase reporter gene) in a 1:1 ratio in serum-free medium. Plasmid DNA was mixed with PEI at a DNA:PEI ratio of 1:3 (3 μg PEI per 1 μg DNA), incubated for 20 min at room temperature, and added to cells dropwise. At 8 h post transfection, the medium was removed, cells were washed with 2 mL of PBS and 2 mL of HEK293T medium with 2.5% FCS were added. At 48 h post transfection, pseudoparticles containing supernatants were harvested by centrifugation for 5 min at 1500 rpm.

**SARS-CoV-2 strains and propagation**. Viral isolates BetaCoV/Netherlands/01/NL/2020 (SARS-CoV-2 D614G variant, #010V-03903), BetaCoV/France/IDF0372/2020 (SARS-CoV-2 WT, #014V-03890) and variant of concern (VOC) Alpha hCoV-19/Netherlands/NH-RIVM-20432/2020 (B.1.1.7, #014V-04031) were obtained from the European Virus Archive global. The VOC Beta 2102-cov-IM-r1-164 (B.1.351) was isolated, sequenced, and kindly provided by Michael Schindler (Tübingen Medical Center, Germany). All strains were propagated on Vero E6 or Caco-2 cells. To this end, 70–90% confluent cells in 75 cm$^2$ cell culture flasks were inoculated with SARS-CoV-2 isolate (multiplicity of infection (MOI) of 0.03-0.1) in 3.5 mL serum-free medium. Cells were incubated for 2 h at 37 °C, before adding 20 mL medium containing 15 mM HEPES. Cells were incubated at 37 °C and supernatant harvested when a strong cytopathic effect (CPE) was visible. Supernatants were centrifuged for 5 min at 1000 × $g$ to remove cellular debris, and then aliquoted and stored at –80 °C as virus stocks. Infectious virus titer was determined as plaque-forming units (PFU) on Vero E6 cells, which was used to calculate MOI.

**Pseudovirus inhibition assay**. Overall, 10,000 Caco-2 cells were seeded in 100 μL DMEM supplemented with 10% FBS, 100 U/mL penicillin, 100 mg/mL streptomycin, 2 mM glutamine, 1× non-essential amino acids, and 1 mM sodium pyruvate. The next day, medium was replaced by 60 μL of fresh medium and cells were treated with 20 μL of serial dilutions of TMPRSS2 inhibitors or small molecule protease inhibitors for 2 h at 37 °C, followed by transduction with 20 μL of infectivity normalized LV(Luc)-CoV-2 pseudoparticles. Transduction rates were assessed after 48 h by measuring luciferase activity in cell lysates with a commercially available kit (Promega). Briefly, cells were washed with PBS and incubated with 40 μL cell lysis reagent for 10 min at room temperature. In all, 30 μL of lysates were transferred to opaque 96-well plates and mixed with 50 μL of Luciferase assay substrate. Luminescence was recorded immediately for 0.1 s/well in an Orion II Microplate luminometer (Berthold) with simplicity 4.2 software. Luciferase activities in absence of inhibitors were set to 100% and $IC_{50}$ were determined by linear regression using GraphPad Prism version 8.4.2.

**SARS-CoV-2 inhibition assay**. Overall, 25,000 Caco-2 cells were seeded in 100 μL respective medium. The next day 40 μL of medium were removed and cells were treated with 20 μL of serial dilutions of TMPRSS2 inhibitors or small molecule protease inhibitors for 2 h at 37 °C, followed by infection with 20 μL SARS-CoV-2 of the respective virus strain at a multiplicity of infection (MOI) of $5 \times 10^{-4}$. Infection rates were assessed at 2 days post infection by in-cell ELISA for SARS-CoV-2 nucleocapsid or spike. Briefly, cells were fixed by adding 180 μL 8% paraformaldehyde (PFA) for 30 min at room temperature and permeabilized by

incubation with 100 μL 0.1% Triton X-100 for 5 min. After washing once with PBS, cells were stained with 1:5000 diluted anti-spike protein antibody 1A9 (Biozol GTX-GTX632604) or anti-nucleocapsid antibody (Sinobiological 40143-MM05) in antibody buffer (10% FBS and 0.3% Tween 20 in PBS) for 1 h at 37 °C. After two washes with 0.3% Tween 20 in PBS, the secondary HRP-conjugated antibody (ThermoFisher #A16066) (1:15,000) was incubated for 1 h at 37 °C. Cells were washed three times with 0.3% Tween 20 in PBS, TMB peroxidase substrate (Medac #52-00-04) was added for 5 min and the reaction stopped using 0.5 M $H_2SO_4$. The optical density (OD) was recorded at 450–620 nm using the Asys Expert 96 UV microplate reader (Biochrom) with DigiRead 1.26 software. Values were corrected for the background signal derived from uninfected cells and untreated controls were set to 100% infection.

**Cytotoxicity assay**. In all, 10,000 Caco-2 cells were seeded in 100 μL respective medium. The next day medium was replaced by 80 μL of fresh medium and cells were treated with 20 μL of serial dilutions of peptidomimetic TMPRSS2 inhibitors or small molecule protease inhibitors. Cell viability was assessed after 48 h with a commercially available kit (Promega). Briefly, medium was removed and cells were lysed with 100 μL CellTiter-Glo reagent for 10 min at room temperature. In total, 50 μL of lysates were transferred to opaque 96-well plates and luminescence was recorded immediately for 0.1 s/well in an Orion II Microplate luminometer (Berthold) with simplicity 4.2 software. Luciferase activities in absence of inhibitors were set to 100%.

**Stability of inhibitors in serum, plasma, and cell culture medium**. The stability of the inhibitors was measured according to a modified procedure from ref. [67]. In short, 10 μL of 10 mM inhibitor solutions were added to 1 mL of 25% (v/v) human serum or plasma in RPMI 1640 and incubated at 37 °C. At indicated timepoints, 100 μL samples were taken and mixed with 200 μL of ethanol to precipitate proteins. The cloudy solution was cooled at 4 °C for 15 min and centrifuged at 14,800 rpm for 2 min. The supernatant was aspirated and analyzed using analytical HPLC. The residual inhibitory constants $K_i$ after incubation was assessed by mixing serial dilutions of the inhibitor with recombinant TMPRSS2 as described above.

**Statistics and reproducibility**. Sample sizes and the number of replicates are indicated in the respective figure legend. The number of experiments is described as $n = X$, triplicates/duplicates describe individual replicates within each experiment.

**Reporting summary**. Further information on research design is available in the Nature Research Reporting Summary linked to this article.

## Data availability
Crystal structures were obtained from Protein Data Bank with accession codes PDB-ID: 1Z8G, 6N4T, 7MEQ, and from SWISS-MODEL repository (https://swissmodel.expasy.org/repository/uniprot/P05981?template=1z8g). Source data are provided with this paper as Supplementary Data 1. Analytical data are presented in Supplementary Fig. 1.

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

## Acknowledgements
This work was supported by grants from the MWK Baden-Württemberg (to J.M.), the EU's Horizon 2020 research and innovation program (Fight-nCoV, 101003555 to J.M.) and by the DFG (CRC1279 to J.M.).

## Author contributions
P.K. and P.M. synthesized the compounds. P.K. and L.W. performed in vitro characterization, stability analyses, and pseudovirus assays. C.K. performed the molecular docking, hit prioritization, and final compound selection for synthesis in agreement with P.K and M.B. T.W., C.C., and J. Müller performed infection assays. P.K., L.W., J.M., and V.M. wrote the manuscript with contributions from all authors. C.K., M.H., S.P., T.S., and K.L. advised and edited the manuscript.

## Funding

## Competing interests
The authors declare the following competing interests: P.K., L.W., J.M., and V.M. filed a patent that claims to use peptidomimetics described herein to inhibit protease activity and cure viral infections. The remaining authors declare no competing interests.
