## [Peer Review File · Communications Biology]

Reviewers' comments:

Reviewer #1 (Remarks to the Author):

In this study, authors developed novel peptidomimetic inhibitors of TMPRSS2, which potently inhibited both pseudotyped and authentic SARS-CoV-2 entry. More importantly, those peptidomimetic TMPRSS2 inhibitors can also broadly inhibit infection by SARS-CoV-2 variants of concern with good off-target selectivity and ideal stability, suggesting that those inhibitors have good potential to be further developed as clinical antiviral agents for prevention and treatment of emerged and emerging SARS-CoV-2 variants.

Overall, this manuscript is well written. The experiments and analyses are technically sound and the methods are sufficiently clear, the results are interpreted appropriately and the conclusions are supported by the data. I did not find any significant weakness, but have a suggestion as shown below.

Recently, new SARS-CoV-2 variants of concern, such as Delta that has become the dominant circulating SARS-CoV-2 variant almost over the world, and Omicron that will soon dominate in many countries. Therefore, authors may assess the inhibitory potency of those inhibitors against infection by the pseudotyped or authentic Delta and Omicron variants.

Reviewer #2 (Remarks to the Author):

Recommendation: major revision

The manuscript by Knaff et al. describes the development and synthesis of peptidomimetic inhibitors of trypsin-like proteases matriptase and TMPRSS2 and demonstrates antiviral activity of a number of the compounds against SARS-CoV-2 pseudotypes and authentic SARS-CoV-2 variants in Caco-2 cells.

The manuscript is well written, the experimental work is clear.

The best candidates, compounds 2, 4 and 5, show potent inhibitory activity against TMPRSS2 and matriptase and suppressed SARS-CoV-2 infection in Caco-2 cells with IC50 values similar to camostat mesylate. Thus, compounds 2, 4 and 5 may provide promising candidates for development of antivirals against SARS-CoV-2 and other respiratory viruses.

However, a weak point of the manuscript is that the authors used matriptase and hespin for molecular modelling and docking studies instead of TMPRSS2. The crystal structure of TMPRSS2 is available since april 2021 (PDB: 7MEQ). Since the major aim of the study is the development of peptidomimetic inhibitors of TMPRSS2, the authors should include docking studies with TMPRSS2 at least with the best candidates.

Another major issue is that the authors do not provide experimental evidence that the trypsin-like activity determined in Caco-2 cells is specifically due to TMPRSS2. Caco-2 cells express several trypsin-like proteases. To strengthen the data the authors should use siRNA knockdown of TMPRSS2 to analyse whether the measured „cell-mediated proteolysis“ is due to TMPRSS2 activity or due to a number of trypsin-like proteases in the cells.

Minor issue: The cell toxicity studies of the compounds (Figure S11) should be implemented in the manuscript and not in the supporting informations.

Reviewer #3 (Remarks to the Author):

The article describes the in silico development and in vitro characterization of peptidomimetic inhibitors of the transmembrane serine protease 2 (TMPRSS2), which plays an important role in

SARS-CoV-2 infection. Binders were identified by molecular docking studies. Peptides were synthesized and coupled to an electrophilic serine trap. The studied lead compounds showed stability in blood serum and plasma for more than 10 days. Selected peptidomimetic inhibitors were demonstrated to block pseudovirus cellular entry as well as an authentic SARS-CoV-2 virus infection. Inhibitory potency was comparable to that of the approved TMPRSS2 inhibitor camostat mesylate. The inhibitors were also active against infection of several SARS-CoV-2 variants of concern. The authors speculate that their inhibitors might be candidates for further preclinical and clinical development not only against SARS-CoV-2 but also against other TMPRSS2-dependent viruses.

This is a very interesting and important report. There are, however, some points the authors ought to address.

Major points:

1. On p22, the authors describe an infection experiment with SARS-CoV-2 isolates from France. They should give some more details about these isolates.
2. On p24, the authors measure the inhibitory potency and stability of TMPRSS2 inhibitors in serum and plasma. What is the inhibitory potency and stability of camostat mesylate under these conditions?
3. If possible, it would be desirable to know the in vivo half-life of the TMPRSS2 inhibitors, again in comparison with camostat mesylate.
4. The authors might want to discuss the usefulness of their inhibitors in comparison with the newly developed Pfizer-inhibitors of the SARS-CoV-2 major protease (Science 2021).

Minor points:

1. Many of the supplemental figures are hard to read or are arranged in a way that the text is not readable.
2. Several references in the list (p29ff) are incomplete

Point-by-point response

Reviewer #1 (Remarks to the Author):

In this study, authors developed novel peptidomimetic inhibitors of TMPRSS2, which potently inhibited both pseudotyped and authentic SARS-CoV-2 entry. More importantly, those peptidomimetic TMPRSS2 inhibitors can also broadly inhibit infection by SARS-CoV-2 variants of concern with good off-target selectivity and ideal stability, suggesting that those inhibitors have good potential to be further developed as clinical antiviral agents for prevention and treatment of emerged and emerging SARS-CoV-2 variants.

Overall, this manuscript is well written. The experiments and analyses are technically sound and the methods are sufficiently clear, the results are interpreted appropriately and the conclusions are supported by the data. I did not find any significant weakness, but have a suggestion as shown below.

Recently, new SARS-CoV-2 variants of concern, such as Delta that has become the dominant circulating SARS-CoV-2 variant almost over the world, and Omicron that will soon dominate in many countries. Therefore, authors may assess the inhibitory potency of those inhibitors against infection by the pseudotyped or authentic Delta and Omicron variants.

We thank the reviewer for appreciating our efforts. We assessed the potency of the compounds 2, 4, 5, 7 against lentiviral pseudotypes of SARS-CoV-2 VOC Delta and Omicron (see below new Fig. 5e and Table S7 of the revised manuscript) and observed no difference in susceptibility towards the inhibitors. We adapted the respective paragraphs accordingly (lines 31-33, 345-346, 559-565, all changes are marked in the revised manuscript).

Figure 5 Peptidomimetic inhibitors reduce SARS-CoV-2 spike driven entry. Peptidomimetic inhibitors and the small molecule camostat mesylate (CM) were added to Caco-2 cells. After 1 h, cells were transduced with lentiviral SARS-CoV-2 pseudoparticles carrying the spike protein of SARS-CoV-2 wildtype (a), Alpha (b), Beta (c), Delta (d), or Omicron BA.1 (e) variant of concern. Transduction rates were assessed 2 days post transduction by measuring luciferase activity in cell lysates. Shown are the means \pm SEM of two independent experiments, each performed in triplicates. Calculated IC₅₀ values for each compound are presented in Table S7.

Table S7 IC₅₀ values of peptidomimetic TMPRSS2 inhibitors and camostat mesylate (CM) against SARS-CoV-2 spike-pseudotyped lentivirus. n.d.: not determined.

Compound	IC ₅₀ [nM]				
	Wuhan-Hu-1	Alpha	Beta	Delta	Omicron BA.1
1	1,201	n.d.	n.d.	n.d.	n.d.
2	1,617	390.5	433.9	627.0	431.0
3	3,243	n.d.	n.d.	n.d.	n.d.
4	1,322	485.7	141.2	610.2	338.8
5	467.2	260.7	153.3	286.1	196.3
6	12,085	n.d.	n.d.	n.d.	n.d.
7	2,068	1,597	920.8	2,367	1,701
8	5,604	n.d.	n.d.	n.d.	n.d.
CM	747.5	98.73	156.4	85.48	77.62

Reviewer #2 (Remarks to the Author):

Recommendation: major revision

The manuscript by Knaff et al. describes the development and synthesis of peptidomimetic inhibitors of trypsin-like proteases matriptase and TMPRSS2 and demonstrates antiviral activity of a number of the compounds against SARS-CoV-2 pseudotypes and authentic SARS-CoV-2 variants in Caco-2 cells. The manuscript is well written, the experimental work is clear. The best candidates, compounds 2, 4 and 5, show potent inhibitory activity against TMPRSS2 and matriptase and suppressed SARS-CoV-2 infection in Caco-2 cells with IC₅₀ values similar to camostat mesylate. Thus, compounds 2, 4 and 5 may provide promising candidates for development of antivirals against SARS-CoV-2 and other respiratory viruses.

We thank the reviewer for appreciating our work.

However, a weak point of the manuscript is that the authors used matriptase and hespin for molecular modelling and docking studies instead of TMPRSS2. The crystal structure of TMPRSS2 is available since april 2021 (PDB: 7MEQ). Since the major aim of the study is the development of peptidomimetic inhibitors of TMPRSS2, the authors should include docking studies with TMPRSS2 at least with the best candidates.

As suggested, we now included the published crystal structure of TMPRSS2 (PDB-ID: 7MEQ) in our docking approach. Comparison of the TMPRSS2 crystal structure with the models confirmed their accuracy (see below, new Fig. S5 of the revised manuscript). Retrospective re-docking experiments with the inhibitors chosen for synthesis revealed binding to the TMPRSS2 crystal structure with favorable positioning of the serine trap moiety (see below, new Fig.1c, Table S3, S4 of the revised manuscript). We adapted the respective paragraphs accordingly (lines 85-86, 106-108, 133-141; 423-463, all changes are marked in the revised manuscript). Altogether, the TMPRSS2 crystal structure confirmed the results obtained from the modelling approach.

Figure S1 Superposition of TMPRSS2 models and crystal structure. TMPRSS2 crystal structure (blue, PDB-ID: 7MEQ, crystallographic ligand in S1 shown with green carbon atoms for orientation), TMPRSS2 homology model (yellow, template hepsin, PDB-ID 1Z8G, C_{α} -RMSD compared to TMPRSS2 is 0.6 Å), matriptase (red, PDB-ID: 6N4T, C_{α} -RMSD compared to TMPRSS2 is 0.6 Å).

Figure 2. Predicted binding of reference binder a) Docking of ace-D-Arg-Pro-Arg-aldehyde reference binder to matriptase surrogate model (white carbon atoms and surface). For clear view, only residues forming polar interactions (yellow dashed lines), and the catalytic residues Ser-195 and His-57 are depicted. b) Docking of ace-D-Arg-Pro-Arg-aldehyde reference binder to hepsin-based TMPRSS2 homology model (white carbon atoms and surface). For clear view, only residues forming polar interactions (yellow dashed lines) and the catalytic residues Ser-186 and His-41 are depicted. c) Docking of ace-D-Arg-Pro-Arg-aldehyde reference binder to TMPRSS2 crystal structure (PDB-ID: 7MEQ, white carbon atoms and surface). For clear view, only residues forming polar interactions (yellow dashed lines), and the catalytic residues Ser-441 and His-296 are depicted. For all panels, carbon atoms of docked ligands are shown in green, oxygen in red and nitrogen in blue. The distance between the nucleophilic serine oxygen and electrophilic carbon atom of the serine trap in angstrom is illustrated by a dashed blue line.

Table S3 Excerpt of docking results of P2-sidechain screening of peptidomimetic inhibitors with ace-D-Arg-X-Arg-aldehyde sequence. A total of 369 molecules were docked. ^a Retrospective docking on the TMPRSS2 crystal structure (PDB-ID: 7MEQ) for peptide sequences selected for synthesis (scores of ketobenzothiazole (kbt)-coupled derivatives **1**, **6**, **7**, **8** in squared brackets). n.d. = not determined. Scores are in kJ/mol. AZc: Azetidine-2-carboxylic acid, Pip: Pípecolinic acid, Pgl: 3,4-Dichlorophenylglycine, Cyc: Cyclobutylalanine. ^b Predicted binding modes are shown in Figure 1a-c.

Ligand sequence ace-D-Arg-X- Arg-Aldehyde	FlexX-score (rank) TMPRSS2 homology model	FlexX-score (rank) Matriptase surrogate	FlexX-score TMPRSS2 ^a [kbt]
Arg	-57.4 (1)	-57.8 (59)	n.d.
Orn	-51.0 (2)	-59.7 (25)	n.d.
Pro ^b	-49.9 (8)	-60.5 (16)	-40.7 [-40.3; 1]
Pgl	-49.2 (9)	-58.8 (41)	n.d.
Phe	-48.4 (15)	-61.2 (9)	n.d.
Lys	-47.5 (23)	-58.6 (45)	n.d.
Thr	-46.9 (29)	-62.2 (4)	-32.7 [-42.5; 8]
Azc	-46.9 (39)	-60.1 (22)	n.d.
Val	-45.9 (54)	-62.1 (6)	n.d.
Pip	-44.1 (101)	-55.9 (110)	-39.6 [-33.2; 6]
Leu	-43.5 (127)	-56.6 (96)	n.d.
Ile	-41.9 (183)	-61.5 (8)	n.d.
Cyc	-39.8 (269)	-52.9 (227)	-32.1 [-35.4; 7]

Table S4 Excerpt of docking results of P3-sidechain screening of peptidomimetic inhibitors with ace-X-Pro/Gly-Arg-aldehyde sequence. For both X-Pro-Arg and X-Gly-Arg sequences 388 molecules were docked and ranked separately. ^a Retrospective docking on the TMPRSS2 crystal structure (PDB-ID: 7MEQ) for peptide sequences selected for synthesis (scores of ketobenzothiazole (kbt)-coupled derivatives **1-5** in squared brackets). Scores are in kJ/mol. n.d. = not determined.

Ligand sequence ace-X-Y-Arg- Aldehyde	FlexX-score (rank) TMPRSS2 homology model		FlexX-score (rank) Matriptase surrogate		FlexX-score TMPRSS2 ^a [kbt]
	Y = Pro	Y = Gly	Y = Pro	Y = Gly	Y = Pro
Arg	-52.0 (1)	-48.3 (3)	-55.1 (298)	-56.4 (28)	-34.3 [-44.0; 2]
D-Arg	-44.2 (135)	-45.5 (10)	-64.9 (1)	-56.4 (27)	-40.7 [-40.3; 1]
His	-39.7 (348)	-40.5 (144)	-54.5 (364)	-54.3 (90)	-35.4 [-41.4; 4]
D-His	-49.3 (6)	-41.8 (92)	-62.2 (16)	-57.4 (6)	-34.3 [-41.9; 3]
Trp	-45.1 (97)	-44.2 (20)	-63.5 (7)	-54,7 (71)	n.d.
D-Trp	-45.1 (104)	-42.8 (50)	-59.7 (70)	53.5 (122)	n.d.
Asn	-40.5 (329)	-40.2 (160)	-52.9 (364)	-56.5 (23)	-36.1 [-42.3; 5]
D-Asn	-46.7 (47)	-42.1 (78)	-57.1 (190)	-51.8 (201)	n.d.
Met	-42.3 (253)	-38.4 (261)	-57.8 (158)	-47.9 (369)	n.d.
D-Met	-38.7 (370)	-35.8 (368)	-59.4 (77)	-50.1 (281)	n.d.
Glu	-39.1 (362)	-40.8 (127)	-54.3 (323)	-50.9 (241)	n.d.
D-Glu	-40.6 (325)	-38.6 (252)	-50.6 (384)	-51.9 (193)	n.d.

Another major issue is that the authors do not provide experimental evidence that the trypsin-like activity determined in Caco-2 cells is specifically due to TMPRSS2. Caco-2 cells express several trypsin-like proteases. To strengthen the data the authors should use siRNA knockdown of TMPRSS2 to analyse whether the measured „cell-mediated proteolysis“ is due to TMPRSS2 activity or due to a number of trypsin-like proteases in the cells.

We agree that this is a critical point to address. As a CRISPR-Cas mediated knockout failed to target all isoforms of TMPRSS2, we performed experiments on HEK293T cells transfected with empty vector or a TMPRSS2 expression plasmid. We observed a dose-dependent reduction of protease activity when incubating TMPRSS2 expressing cells with our inhibitors, even after correcting the signal using mock-transfected HEK293T cells. (see below, Fig. 4b and Table S6 of the revised manuscript). The reduction in cellular protease activity was 15-20× more pronounced in TMPRSS2 expressing cells than in mock-transfected cells. Thus, the inhibitors indeed reduce the activity of cellular TMPRSS2, as expected from the experiments with the recombinant enzyme. We adapted the respective paragraphs accordingly (lines 328-337; 529-538, all changes are marked in the revised manuscript).

Figure 4 Peptidomimetic inhibitors block cellular protease activity. a) Peptidomimetic compounds 2, 4, 5 and 7 as well as camostat mesylate (CM) and FOY-251 were added to Caco-2 cells. After 30 min, the fluorogenic reference substrate Boc-Gln-Ala-Arg-AMC was added, and the reaction rate of substrate degradation was assessed by recording the fluorescence intensity within 2 h. b) peptidomimetic compounds 2, 4, 5 and 7 as well as camostat mesylate (CM) were added to HEK293T cells transiently expressing TMPRSS2, followed by addition of fluorogenic reference substrate Boc-Gln-Ala-Arg-AMC. Graph shows normalized fluorescence intensities after 2 h, corrected for the signal of mock-transfected HEK293T cells. Shown are the means \pm SD of triplicate (a) or duplicate (b) measurements. Calculated IC_{50} values for each compound are presented in Table S6.

Table S6 IC_{50} values of peptidomimetic TMPRSS2 inhibitors, camostat mesylate (CM) and FOY-251 measured on Caco-2 or TMPRSS2 expressing HEK 293T cells. n.d.: not determined.

Compound	IC_{50} Caco2 [nM]	IC_{50} HEK293T TMPRSS2 [nM]
2	32	3.5
4	45.5	13.3
5	12.7	2.2
7	234.2	27.9
CM	42.2	1.1
FOY-251	377.2	n.d.

Minor issue: The cell toxicity studies of the compounds (Figure S11) should be implemented in the manuscript and not in the supporting informations.

The toxicity data is now displayed in Fig. 6 e) (see below) of the revised manuscript.

Figure 6 Peptidomimetic inhibitors reduce SARS-CoV-2 infection. Peptidomimetic inhibitors and the small molecule camostat mesylate (CM) were added to Caco-2 cells. After 1 h, cells were infected with SARS-CoV-2 Wuhan-Hu1 (a), SARS-CoV-2 bearing the spike D614G mutation (b), or the variants of concern Alpha (c) and Beta (d). Infection rates were determined 2 days post infection by in cell ELISA for the viral N protein (e) Cytotoxicity of peptidomimetic inhibitors. Inhibitors and the small molecule camostat mesylate (CM) were added to Caco2 cells. Cell viability was assessed 2 days post addition by measuring ATP content in cell lysates. Due to low stock concentration compounds 1, 6 and 8 were tested at a maximum concentration of 20,000 nM. Shown are the means \pm SEM of three independent experiments (a-d) or mean \pm SD of one experiment (e), each performed in triplicates. Calculated IC₅₀ values for each compound are listed in Table S8.

Reviewer #3 (Remarks to the Author):

The article describes the in silico development and in vitro characterization of peptidomimetic inhibitors of the transmembrane serine protease 2 (TMPRSS2), which plays an important role in SARS-CoV-2 infection. Binders were identified by molecular docking studies. Peptides were synthesized and coupled to an electrophilic serine trap. The studied lead compounds showed stability in blood serum and plasma for more than 10 days. Selected peptidomimetic inhibitors were demonstrated to block pseudovirus cellular entry as well as an authentic SARS-CoV-2 virus infection. Inhibitory potency was comparable to that of the approved TMPRSS2 inhibitor camostat mesylate. The inhibitors were also active against infection of several SARS-CoV-2 variants of concern. The authors speculate that their inhibitors might be candidates for further preclinical and clinical development not only against SARS-CoV-2 but also against other TMPRSS2-dependent viruses.

This is a very interesting and important report. There are, however, some points the authors ought to address.

We thank the reviewer for acknowledging our work.

Major points:

1. On p22, the authors describe an infection experiment with SARS-CoV-2 isolates from France. They should give some more details about these isolates.

We rephrased the text and replaced the expressions “B.1.1.7”, “B.1.351” etc. by the official VOC designations “Alpha”, “Beta” etc. so that the isolates/spike proteins can be clearly assigned (line 100-101, 345-346, 354-359, 559-565, 589-592, all changes are marked in the revised manuscript).

2. On p24, the authors measure the inhibitory potency and stability of TMPRSS2 inhibitors in serum and plasma. What is the inhibitory potency and stability of camostat mesylate under these conditions?

To directly address this question, we performed extensive studies on the stability/activity of our inhibitors and camostat mesylate (CM) in serum and plasma, and show the data in the revised manuscript (see below new Figure 7, and new Figs S13 and S14). In plasma and serum, compounds 2 and 7 largely retained their activity against TMPRSS2 over this period of time, whereas CM showed a ~60-fold reduced potency immediately after addition to body fluids. We adapted the respective paragraphs accordingly (lines 408-415, 617-631, 674, 676-677, all changes are marked in the revised manuscript).

Figure 7 Serum stability of inhibitors. Compound 2 (a), compound 7 (b), or camostat mesylate (c) were incubated in human serum for indicated timepoints. Samples were mixed with recombinant TMPRSS2, followed by the addition of the fluorogenic reference substrate BOC-Gln-Ala-Arg-AMC. Graph shows normalized fluorescence intensities after incubation for 2 h. Negative ctrl: no inhibitor, positive ctrl: inhibitor in assay buffer. d) Inhibitory constants K_i as determined from (a-c). Shown are the means \pm SD of triplicate measurements. No inh.: no inhibition.

Figure S13 Plasma stability of inhibitors. Compound 2 (a), compound 7 (b), or camostat mesylate (c) were incubated in human plasma for indicated timepoints. Samples were mixed with recombinant TMPRSS2, followed by the addition of the fluorogenic reference substrate BOC-Gln-Ala-Arg-AMC. Graph shows normalized fluorescence intensities after incubation for 2 h. Negative ctrl: no inhibitor, positive ctrl: inhibitor in assay buffer. d) Inhibitory constants K_i as determined from (a-c). Shown are the means \pm SD of triplicate measurements. No inh.: no inhibition.

Figure S14 Cell culture medium stability of inhibitors. Compound 2 (a), compound 7 (b), or camostat mesylate (c) were incubated in cell culture medium for indicated timepoints. Samples were mixed with recombinant TMPRSS2, followed by the addition of the fluorogenic reference substrate BOC-Gln-Ala-Arg-AMC. Graph shows normalized fluorescence intensities after incubation for 2 h. negative ctrl: no inhibitor, positive ctrl: inhibitor in assay buffer. d) Inhibitory constants K_i as determined from (a-c). Shown are the means \pm SD of triplicate measurements. No inh.: no inhibition.

3. If possible, it would be desirable to know the *in vivo* half-life of the TMPRSS2 inhibitors, again in comparison with camostat mesylate.

We understand that it is important to assess the *in vivo* half-life of our inhibitors in the pre-clinical development. Even though *in vivo* studies will be conducted in the future, they are beyond the scope of this article. The data shown in Figure 7 and S13, S14 of the revised manuscript are promising in this regard. The *in vivo* half-life of GBPA - the active metabolite of camostat mesylate - has been determined to be 60 – 100 min. after intravenous administration (doi.org/10.3109/00498259409043223, https://www.bfarm.de/SharedDocs/Downloads/DE/Arzneimittel/Zulassung/amInformationen/Lieferengpaesse/Uebersetzung_FOIPAN.pdf?__blob=publicationFile).

4. The authors might want to discuss the usefulness of their inhibitors in comparison with the newly developed Pfizer-inhibitors of the SARS-CoV-2 major protease (Science 2021).

We have now mentioned the new approved direct acting antiviral drugs in the introduction. We also discuss the advantage of targeting a cellular enzyme with regards to resistance development in the discussion (lines 55-58; 680-682), all changes are marked in the revised manuscript).

Minor points:

1. Many of the supplemental figures are hard to read or are arranged in a way that the text is not readable.

We restructured the supplement to make the figures more readable and to improve the flow of the main text. We adapted the respective paragraphs accordingly (all changes are marked in the revised manuscript).

2. Several references in the list (p29ff) are incomplete

The reference list was updated in the revised manuscript.

REVIEWERS' COMMENTS:

Reviewer #1 (Remarks to the Author):

The authors have completed new experiments suggested, provided satisfactory explanation to the questions raised, and revised the manuscript accordingly. I thus recommend the acceptance of this manuscript for publication in Communications Biology.

Point-by-point response

Reviewer #1 (Remarks to the Author):

The authors have completed new experiments suggested, provided satisfactory explanation to the questions raised, and revised the manuscript accordingly. I thus recommend the acceptance of this manuscript for publication in Communications Biology.

We thank the reviewer for suggestions to the manuscript and for the recommendation for publication.